# Integrated single-cell atlas of human atherosclerotic plaques

Korbinian Traeuble [1,2,3], Matthias Munz [3], Jessica Pauli [4,5], Nadja Sachs [5,6], Eshan Vafadarnejad[3], Tania Carrillo-Roa [3], Lars Maegdefessel [4,5], Peter Kastner [3,7] ✉ & Matthias Heinig [1,2,5,7] ✉

Atherosclerosis, a major cause of cardiovascular diseases, is characterized by the buildup of lipids and chronic inflammation in the arteries, leading to plaque formation and potential rupture. Despite recent advances in single-cell transcriptomics (scRNA-seq), the underlying immune mechanisms and transformations in structural cells driving plaque progression remain incompletely defined. Existing datasets often lack comprehensive coverage and consistent annotations, limiting the utility of downstream analyses. Here, we present an integrated single-cell atlas of human atherosclerotic plaques, covering roughly 250k high-quality annotated cells. We achieve robust cell type annotations validated by expert consensus and surface protein measurements. Using this atlas, we introduce distinct markers for plaque neutrophils, identify a proangiogenic endothelial cell cluster enriched in advanced lesions, and specialized macrophage subsets. We also establish that fibromyocytes are exclusive to vascular tissue. This comprehensive atlas enables accurate automatic cell type annotation of new datasets, improves experimental design by guiding sample size and detection power, and supports the deconvolution of bulk RNA-seq data. An interactive WebUI makes these resources widely accessible.

Atherosclerosis is the primary pathology behind acute ischemic cardiovascular events, like myocardial infarction and stroke[1]. It is characterized by lipid accumulation and chronic inflammation in the arteries, leading to plaque formation and potential rupture[2,3]. The overarching immune mechanisms and transformation and differentiation processes in vascular cells that reside within the affected arteries, such as vascular smooth muscle cells (SMCs), endothelial cells (ECs), and fibroblasts involved in plaque progression are incompletely understood and under current investigation. The recent advances of single-cell transcriptomics (scRNA-seq) gives relevant insights into these processes, unraveling previously undetermined roles of immune and non-immune cells. For example, Wirka et al.[4] characterized modulated SMCs that transform

into fibroblast-like cells they termed fibromyocytes, which play a protective role in coronary artery disease. Fernandez et al.[5] thoroughly investigated the contribution of T cells and macrophages, and identified certain subsets associated with plaque vulnerability.

Many public scRNA-seq datasets of human atherosclerotic plaques are available, but often do not cover the full breadth of disease-contributing cell types, as they are focused on specific subtypes based on cell sorting approaches and preprocessing being applied prior to library preparation and sequencing[6–9]. Additionally, some cell types are easier to harvest in scRNAseq, and hence more abundant in the datasets[10]. Currently, cell type annotations of many of these data sets are not publicly available, making the annotation of cell types and

[1]Institute of Computational Biology, German Research Center for Environmental Health, Helmholtz Zentrum München, Neuherberg, Germany. [2]Department of Computer Science, TUM School of Computation, Information and Technology, Technical University of Munich, Garching, Germany. [3]Research & Early Development, Discovery Sciences, Roche Diagnostics GmbH, Penzberg, Germany. [4]Institute of Molecular Vascular Medicine, TUM University Hospital Rechts der Isar, Munich, Germany. [5]German Center for Cardiovascular Research, partner site Munich Heart Alliance, Berlin, Germany. [6]Department for Vascular and Endovascular Surgery, TUM University Hospital Rechts der Isar, Munich, Germany. [7]These authors contributed equally: Peter Kastner, Matthias Heinig. ✉e-mail: peter.kastner@roche.com; matthias.heinig@helmholtz-munich.de

composition of plaques a major challenge that requires expert knowledge. Because most downstream analysis tasks require cell type information, accurate consensus annotation of cell types across datasets is of utmost importance. Consequently, single-cell atlases that integrate and harmonize various published datasets, such as the human lung cell atlas[11] or the heart cell atlas[12], emerged as useful references enabling coherent downstream analyses[13], such as automatic cell type annotation of new datasets[11], optimal experimental design[14], interpretation of genetic association studies[11,15,16] and deconvolution of bulk RNA-seq data sets[17]. These atlases typically comprise a comprehensive integrated data resource along with a model trained on this data. This combination allows for accurate data annotation and promotes re-use[18]. Atlases of different tissues are ultimately paving the way towards a human cell atlas[19] that can be used to train foundation models[20–22], which require vast amounts of data.

While a comprehensive understanding of atherogenesis requires insights into both healthy and diseased arterial cells, obtaining truly "healthy" samples remains a significant challenge. Studies on "healthy" arteries often involve patient populations with underlying cardiovascular conditions, such as end stage heart failure[23] or earlier stages of atherosclerosis[24–26], making it difficult to establish a definitive "normal" phenotype. To address this limitation, we focused our analysis on characterizing the cellular heterogeneity across various stages of atherosclerotic plaques. To set the stage for future comparisons, it is key to obtain a high-resolution cell type annotation of the cells of atherosclerotic plaques.

First single-cell atlases of atherosclerotic lesions[27,28] have already been proposed. These atlases have several shortcomings, as they are composed mainly of non-atherosclerotic tissue samples, inflating the number of cells in the atlas. As a consequence, the effective number of plaque specific cells is still relatively low, which limits the robustness of cell type annotations and the ability to detect rare but disease-relevant cell types. Moreover, there is currently no evaluation of the consensus of annotations in the field. Finally, existing atlases are limited to two types of arteries, carotids and coronaries, neglecting plaques from other arterial sites like femoral arteries of great relevance to vascular occlusive disease (PVOD). A key practical limitation appears to be the lack of publicly available annotations and model weights to effectively make use of the existing atlases.

For this current study, we have curated an easily accessible plaque cell atlas that encompasses the most comprehensive dataset to date

with 259,493 high quality annotated cells from human atherosclerotic carotid, coronary and femoral arteries. We applied the best performing data integration method, selected from a wide spectrum of available models through the most comprehensive model benchmark on a range of metrics specifically evaluated on plaque single-cell data sets. The cell type annotations were orthogonally validated with expert annotation consensus and surface protein measurements. We made the annotations and model weights easily accessible by providing an easy-to-use interactive WebUI to automatically annotate new datasets including uncertainty. The performance of the atlas and model was demonstrated and validated in several downstream tasks, such as automated cell type annotation, planning of future experiments with scPower[14], and combining the vast information of single-cell data with the big sample sizes of bulkRNA-seq datasets by deconvolution with BayesPrism[17]. Overall, this comprehensive and robust atlas marks a significant step forward in understanding the complex mechanisms of atherosclerosis and enhances the utility of transcriptomic profiling technologies in cardiovascular research.

## Results

### Integration of public datasets into one atlas

We collected all publicly available single-cell datasets of plaque from carotid, coronary and femoral arteries covering a total of 259,493 cells (after quality control (QC)) of diverse cell types and pathologies (see Table 1). Fig. 1 shows the workflow to construct the single-cell plaque atlas. First, we pre-processed all datasets as described in the original publications. If no detailed description was available, we applied our own dataset-specific quality metrics (see "methods" section for details). Subsequently, we applied ambient RNA correction[29] and doublet detection[30] on each dataset/sample independently.

Next, the datasets were integrated into one latent space to eliminate technical batch effects and conserve biological signals. To select the best method for this task, we performed an extensive benchmark with scib-metrics[31] of commonly used integration methods: scVI[32], Harmony[33], LIGER[34], scANVI[35], scGen[36], scPoli (with negative binomial and mean squared error loss)[37] and *baseline PCA*. The benchmark was evaluated on ten metrics in the categories batch correction and bio conservation, where the latter requires cell type labels. For this reason and following a guideline for atlas curation[18], we manually annotated a subset of 11 samples using a carefully curated table of human plaque-specific marker genes (see Fig. 2a). This table, the most comprehensive to date, includes genes previously utilized for annotating plaque cells in scRNA-seq studies. The 11 samples were manually annotated incrementally until the total number of cells in each cell type was at least 600 (see Supplementary Fig. 1). This enabled us to achieve robust annotations and to cover all major cell types present in the subset of samples, which we designated as "level 1" annotation. Next, we compared methods based on these annotations and other metrics. As expected, the baseline PCA had the worst batch correction score out of all methods (see Supplementary Fig. 2). Surprisingly, scVI and LIGER performed poorly in bio conservation, while having a high batch correction score, indicating an over correction. Fine tuning the scVI model with the scANVI method demonstrated a substantial improvement in the bio conversation score. Overall, the method scPoli outperformed all other methods in both bio conservation and batch correction metrics and was hence used to integrate all the remaining samples as well.

We then proceeded with the atlas building step by transferring level 1 cell type annotations to the remaining unlabeled cells using the scArches[38] method. Subsequently, we examined each level 1 cell type cluster to manually adjust potentially mis-annotated cells. For instance, we identified subclusters expressing T cell specific markers among cells initially annotated as natural killer (NK) cells at level 1; these cells were consequently relabeled as T cells. Notably, we observed a distinct cluster within the monocyte population where typical monocyte markers were not expressed. This cluster

**Table 1 | All publicly available scRNA-seq datasets for plaque tissues in humans from coronary, femoral and carotid arteries in the atlas after excluding low quality, uncertain cells and undefined cells in level 2 annotations**

| Dataset | Plaque location | # Cells | # Samples | Accession number |
|---|---|---|---|---|
| Pan et al.[70] | carotid | 8850 | 3 | GSE155512 |
| Alsaigh et al.[26] | carotid | 34,626 | 3 | GSE159677 |
| Ahmad et al.[48] | carotid | 5554 | 2 | GSE179159 |
| Dib et al.[8] | carotid | 25,147 | 6 | GSE210152 |
| Fernandez et al.[5] | carotid | 9935 | 7 | GSE224273 |
| Slysz et al.[9] | carotid | 27,245 | 3 | GSE234077 |
| Pauli et al.[25] | carotid | 1269 | 8 | GSE247238 |
| Bashore et al.[47] | carotid | 75,582 | 18 | GSE253904 |
| Wirka et al.[4] | coronary | 11,750 | 8 | GSE131778 |
| Emoto et al.[6] | coronary | 1621 | 2 | GSE184073 |
| Chowdhury et al.[7] | coronary | 22,543 | 12 | GSE196943 |
| Slysz et al.[9] | femoral | 35,371 | 7 | GSE234077 |
| Total | | 259,493 | 79 | |

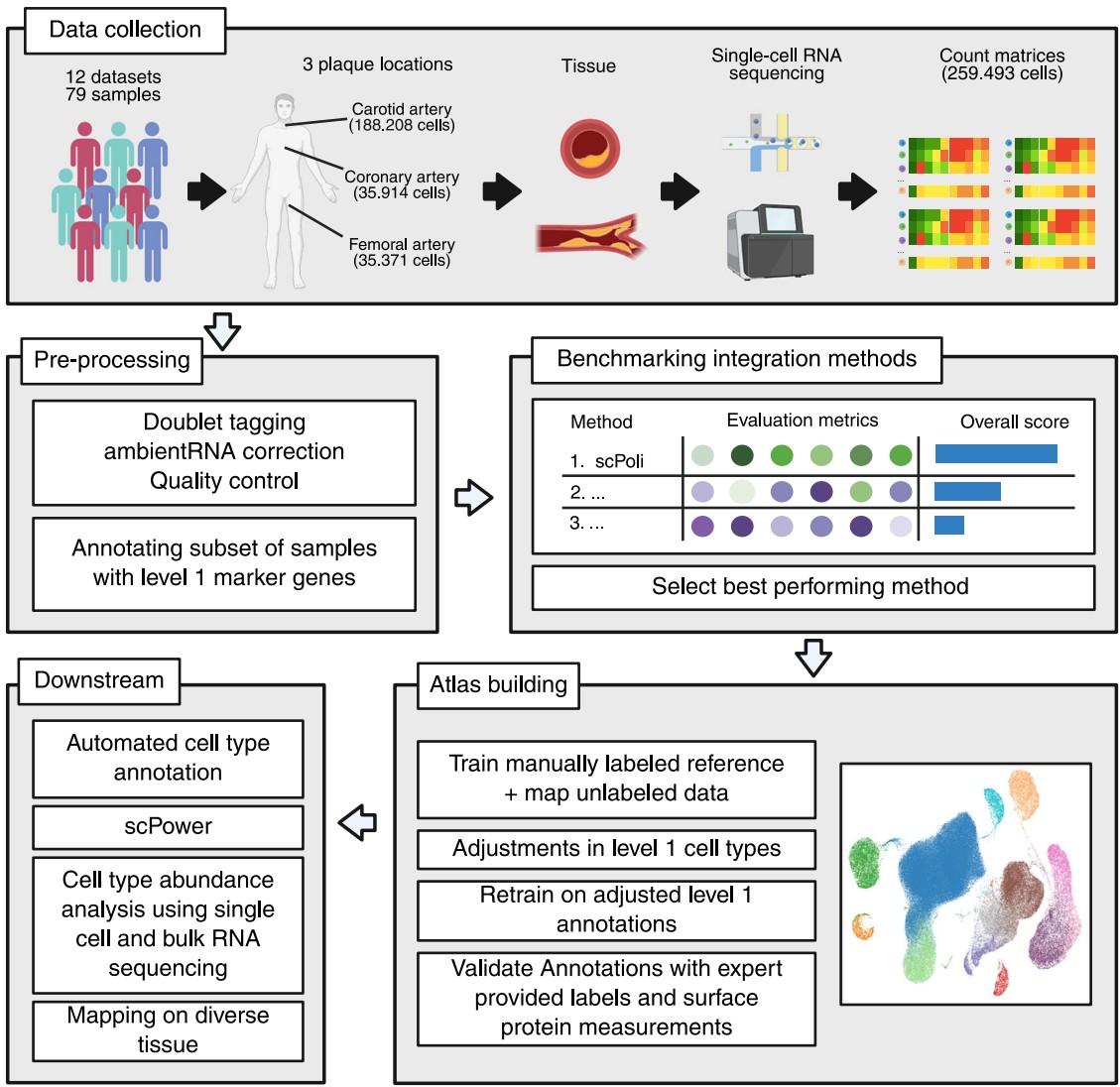

**Fig. 1 | Workflow to construct the single-cell plaque atlas.** (1) All publicly available scRNA-seq datasets from atherosclerotic plaques from carotid, coronary and femoral arteries were collected from the NCBI GEO database. (2) The same preprocessing pipeline was applied to all datasets. Doublets were tagged, counts were corrected for ambient RNA and cells and genes were filtered as described in the original publication. A subset of samples was manually annotated with level 1 marker genes for the subsequent benchmark of integration methods. (3) A benchmark of integration methods on five batch correction and five biological conservation metrics identified the best performing method, scPoli. (4) ScPoli was applied on the subset of annotated samples to build the atlas. Subsequently, we made adjustments on the level 1 cell types, retrained the reference and mapped the yet unclear cells. The resulting atlas was validated by expert provided labels and surface protein measurements. (5) The annotated plaque atlas was used in the following downstream applications: First, automatic cell type annotation, which was validated with an independent dataset (Bashore et al.) and included in the subsequent reference. This reference was subclustered to define finer-grained level 2 cell types, which were then used for bulk deconvolution and abundance analysis. Lastly, the atlas was used to design future experiments with scPower and mapped on non-lesion arteries and diverse tissue compendium. Created in BioRender. Träuble, K. (2025) https://BioRender.com/keay6f9.

consistently appeared regardless of the batch correction technique employed. It distinctly expressed neutrophil-associated genes such as *NAMPT*[39], *IFITM2*[40], *GOS2*[41], *CXCL8*[42], *NEAT1*[41], *SRGN*[43] and *AQP9*[44], suggesting that these cells are neutrophils. This finding is particularly interesting because neutrophils are known to be difficult to detect in scRNAseq[45]. This challenge may be attributed to their high content of readily releasable ribonucleases that rapidly degrade endogenous RNA and their limited gene expression profile, consisting of only a few hundred genes[46]. In line with that, 10x Genomics, the manufacturer of the scRNAseq technology used, indicates that the Cell Ranger software may, by default, filter out neutrophils. In plaque tissue, only one study[27] has detected neutrophils in very low numbers, and in the Bashore et al. study[47] they were identified using surface proteins.

All uncertain cells were labeled as uncertain and subsequently mapped on the retrained corrected reference. SMCs, fibromyocytes,

and fibroblasts form a distinct cluster, as do macrophages, DCs, and monocytes. Similarly, T cells and NK cells are grouped together, while B cells, plasma cells, and ECs all form their own separate clusters (Supplementary Fig. 3). The dot plot depicting the marker genes expression in this reference atlas is shown in Fig. 2b.

To validate the cell type annotations of the atlas, we used two orthogonal methods. First, via label consensus, and secondly with unbiased protein surface markers from CITE-seq data. Three of the datasets (Pauli[25], Emoto[6] and Wirka[4]) were annotated independently by an expert and annotations of one dataset were provided by the authors (Ahmad et al.[48]). The predicted cell types in our atlas were compared to the provided labels. The precision of the annotations is 89.79% in the Pauli dataset and higher than 90% in the three other datasets (see Fig. 3a–d). The confusion matrices also show the mismatch of labels mostly within cell types that are hard to distinguish because of similar

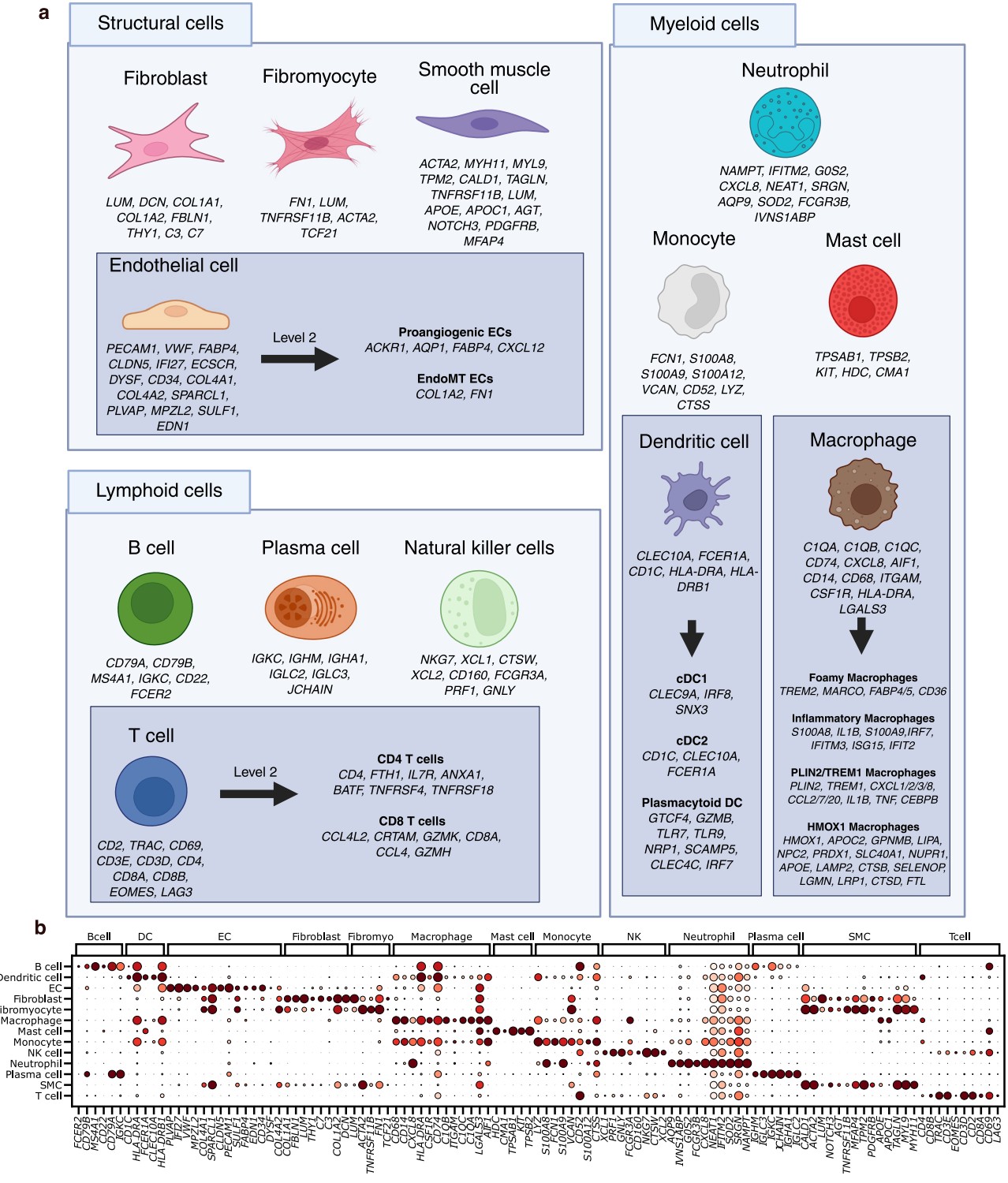

**Fig. 2 | Cell type annotation. a** Shows an overview of marker genes in atherosclerotic plaques previously used in scRNA-seq datasets to annotate cells. Level 1 cell types refer to major cell types commonly used in studies, while level 2 refers to a finer grained sublevel annotation of specific cell types with a distinct biological function. A more detailed list of references and the cell types' role in the disease progression are provided in Supplementary Table 1. **b** Shows a dot plot of the level 1 marker genes in the predicted cell types in the reference atlas. The color coding shows normalized gene expression of the marker genes (y-axis) in each cell type (x-axis), with light red indicating low expression and dark red indicating high expression. Genes are shown repeatedly if they serve marker genes for multiple cell types. Created in BioRender. Träuble, K. (2025) https://BioRender.com/2mwy4mj.

transcriptomes, such as NK and T cells or DC, macrophages and monocytes. SMCs, fibroblasts and fibromyocytes are also more difficult to distinguish for the model, and they form one cluster in the UMAP. Overall the consensus is very high with residual uncertainty only between closely related cell types, demonstrating the robustness of the annotations. These results demonstrate a very high consensus between different manual annotations, but it appears impossible to decide which annotation is more accurate in the absence of objective ground truth cell type labels. For a more unbiased validation, we analyzed CITE-seq data[5] of a sample included in the atlas. The mRNA

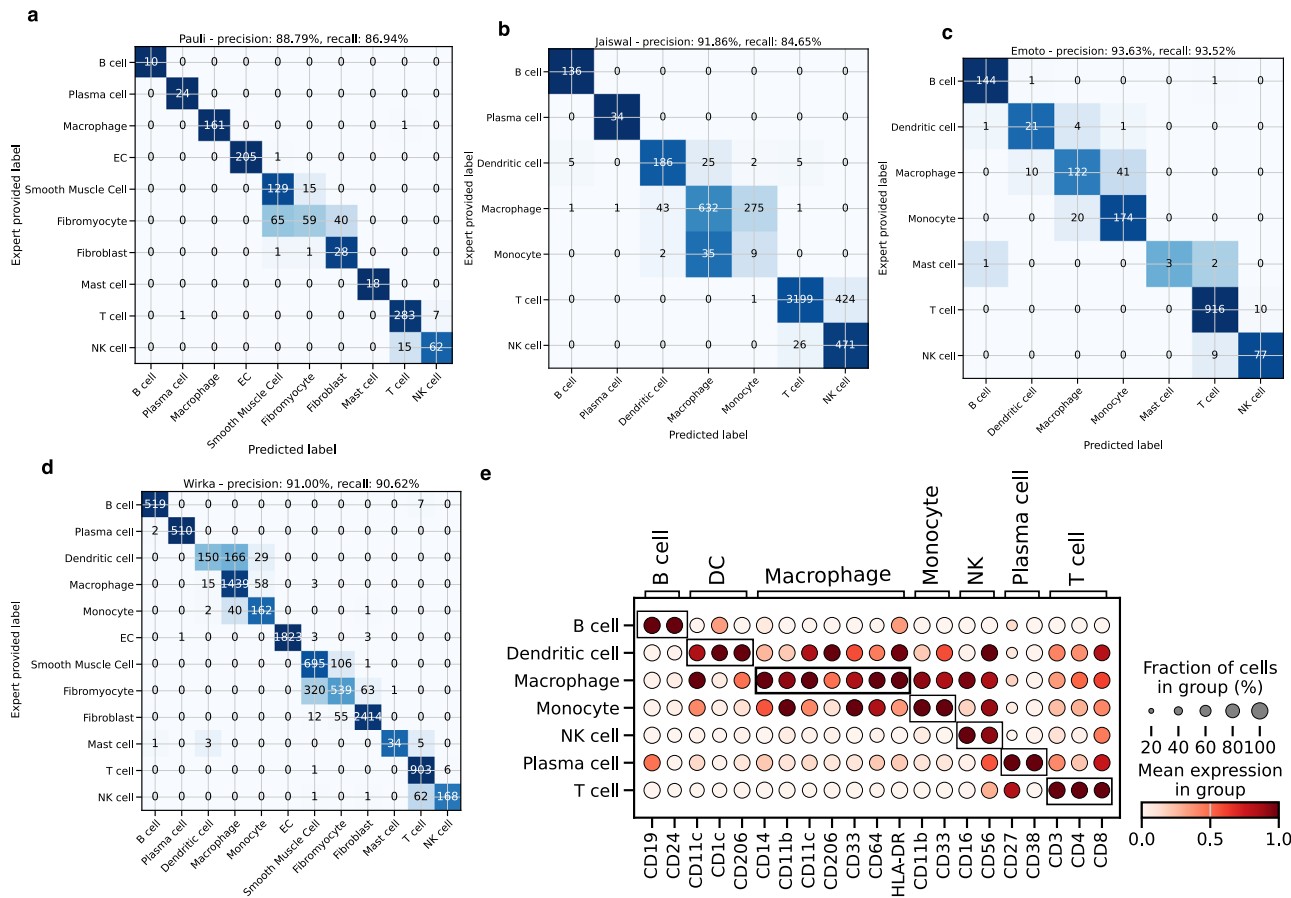

**Fig. 3 | Validation of cell type label transfer. a–d** show the confusion matrices for different data sets (data set name given in the panel headers) providing the number of cells with specific combinations of expert annotations ("Expert provided Label" on the y-axis) and labels predicted by the atlas ("Predicted Label" on the x-axis). To compensate for the differences in cell type abundance the colormap is normalized along the rows and the overall precision and recall are weighted according to their abundance. **e** Depicts the dot plot of surface markers grouped according to cell types (x-axis) in a CITE-seq sample to orthogonally validate the cell type annotations of the atlas (y-axis). The highly specific expression pattern of the surface protein is depicted with black boxes. The color coding shows normalized protein expression of the proteins encoded by the marker genes, with light red indicating low expression and dark red indicating high expression. Source data are provided as a Source Data file.

expression data of these cells was part of the atlas, and was used to predict cell type labels. These predicted labels were compared to the proteomics surface markers, which were not used to make predictions. The available surface markers were grouped by their cell type specificity and visualized across our cell type annotations (see Fig. 3e). The surface markers show highly specific expression patterns across predicted cell types, providing an unbiased validation of our cell type annotation on the protein level.

Overall, this extensive and thorough orthogonal validation demonstrates the robustness of the annotations in the atlas and provides confidence in using it as a reference for future studies.

**Automatic cell type annotation**

A key use case of the plaque cell atlas is to automatically annotate and integrate future scRNA-seq plaque datasets by reference mapping. To validate the quality of automatic cell type annotations based on our atlas, we analyzed an additional independent carotid plaque dataset[47] by Bashore et al. consisting of more than 75k cells, of which more than 25k have additionally been profiled by CITE-seq. We utilized this as a query data set and mapped it to the atlas as a reference using scArches. Fig. 4 shows the UMAP projection of the query onto the reference colored by level 1 (Fig. 4a, b) cell type annotation. In addition to the annotations, the uncertainty of the cell type assignments is provided,

which is derived from the scaled euclidean distance to the closest cell type prototype in the reference[37] (see Fig. 4c). The annotation shows strong confidence in the majority of cells, while only a small percentage of cells have a high uncertainty. The UMAP projections of the whole atlas including the automatically annotated cells shows that cell types of query cells were assigned consistently with the reference (Fig. 4d). Cells with uncertainty greater than 0.5 were excluded (1449 cells out of 77,112). This atlas is used for all subsequent level 1 downstream tasks.

To demonstrate the consensus of annotations within the field, we compared our predictions with the annotations provided by the authors (see Fig. 4e). In a first analysis, the authors' annotations were harmonized to match our cell types, keeping only the common cell types. Our predictions achieved a precision of 94.01% and a recall of 91.74%, demonstrating high consensus. Similar to the atlas validation, blocks between transcriptionally similar cell types are forming. Notably, while Bashore et al. had used surface protein expression to annotate neutrophils, we obtained almost perfect precision and recall for neutrophil annotations only based on transcriptomic profiles.

To corroborate the accuracy of predicted cell types, we assessed the expression levels of surface proteins measured by CITE-seq. The dot plot indicates highly cell type specific expression of established surface protein markers[47] for all cell types (Fig. 4f), indicating a high

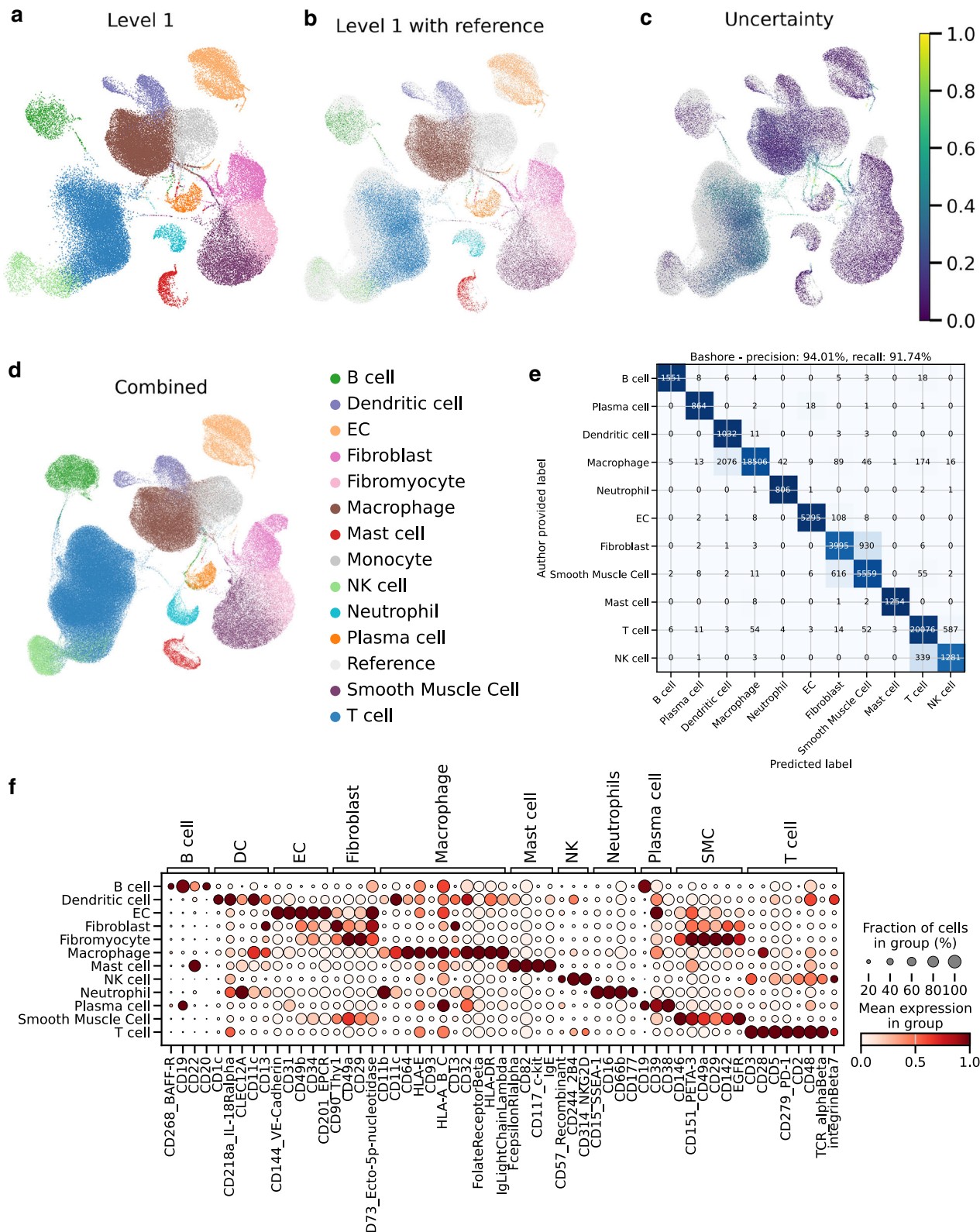

**Fig. 4 | Reference mapping with a dataset consisting of more than 75k cells.**
**a** the UMAP projection of the query cells with cells colored by their annotations on level 1 while (**b**) additionally includes cells of the reference in gray. The uncertainties of the predictions are shown by the color code in (**c**). **d** The whole atlas, including the colored reference. **e** Confusion matrix comparing labels (y-axis) provided by the authors of the original datasets with our predictions (x-axis) with a row normalized colormap to account for differences in cell type abundances. **f** Dot plot of CITE-seq protein expression of surface markers grouped according to cell types (x-axis) for each annotated cell type (y-axis). The highly specific expression pattern of the surface protein is depicted with black boxes. The color coding shows normalized protein expression of the proteins encoded by the marker genes, with light red indicating low expression and dark red indicating high expression. Cells with uncertainty greater than 0.7 are excluded from this analysis. Source data are provided as a Source Data file.

agreement of predictions with the cells' true identity. Importantly, fibromyocytes express both SMC and fibroblast markers, as these cells are rendered fibroblast-like SMCs[4]. Additionally, the neutrophils, annotated based on our mRNA derived markers, express known neutrophil surface proteins CD15 and CD16. Together with the consensus to the author-provided annotation, this highlights the robustness of our proposed marker genes.

This validation by unbiased CITE-seq and annotation consensus confirms that the plaque atlas yields highly accurate automatic cell type annotations. Additionally, we investigated the annotations of cell types that were not shared between our dataset and the authors' dataset. In the Bashore et al. dataset, monocytes and fibromyocytes were not identified. As expected, our predicted fibromyocytes were classified as fibroblasts and SMCs by the authors, while our monocytes were determined mostly as macrophages (Supplementary Fig. 4). Cells that were not annotated by the authors were mostly labeled as macrophages in our analysis.

To enhance accessibility for researchers annotating their datasets, we provide our atlas as a reference via the user-friendly archmap web server and also offer a solution that does not require data upload, comprising a preconfigured Python script and a Docker container. Both approaches eliminate the need for advanced technical proficiency, enabling researchers of any background to efficiently annotate their datasets. Having established a validated reference for plaque cell-type annotation, we next asked how the atlas could inform the prospective design of future studies. Powered sample-size estimation is a critical yet often overlooked step in single-cell experiments, particularly in human cardiovascular research where tissue availability and sequencing budgets are limited.

## Atlas-guided experimental design with scPower

Leveraging the cell-type-specific gene-expression priors contained in the plaque atlas, we applied the scPower[14] framework to estimate the sample sizes, cell numbers and sequencing depth required to detect biologically meaningful differential-expression signatures across plaque cell populations. This requires assumptions about the gene expression levels and effect sizes (log fold change) of genes of interest. To enable plaque-specific power analysis, we used the atlas data to learn the parameters of these cell type specific gene expression prior distributions (Supplementary Fig. 5). Moreover, we derived several technical parameters from the Pauli et al.[25] dataset, which are required as additional input. In the analyses that follow, we specified a fixed budget as well as the costs of typical sequencing runs (see "Methods").

To evaluate the power to detect differentially expressed gene signatures, we formulated assumptions for three scenarios with different average effect sizes of differential gene expression: a pessimistic, neutral, and optimistic scenario with fold changes (FC) 1.1, 1.5 and 2.5 respectively. ScPower can estimate power for specific gene sets, which reflect the aims of the experiment and can be derived from prior knowledge. Here, we made use of this feature and evaluated the average power for three atherosclerosis-related gene sets, which are expected to be active in different cell types: (1) targets of the CCL19[49] chemokine, which is overexpressed in carotid plaques of symptomatic patients[50,51] and its targets downstream of CCR7 are expected to be expressed in T-cells, B-cells, dendritic cells and NK cells[52,53]; (2) genes of the Interferon-Gamma (IFN-γ) pathway, which is active in atherosclerotic lesions and a well known activator of macrophages[54]; and (3) genes of the vascular remodeling pathway, which were identified through genome-wide association studies of coronary artery disease[55] and are expected to be active in structural cells. For all combinations of these specific pathways and fold changes, we evaluated the power to identify differentially expressed genes with a total sample size of 22 individuals.

For abundant cell types the power is generally high, while lower abundance cell types show substantially less power (Fig. 5). As expected, higher effect size leads to an increase in power. For detecting differential expression in the vascular remodeling pathway, structural cell types, such as SMCs, ECs, fibromyocytes, and fibroblasts, exhibit comparatively higher statistical power than other cell populations, despite their lower abundances, as the relevant genes are highly expressed in these cells. Accordingly, the IFN-γ signaling pathway demonstrates its strongest power in immune cells, including T cells, macrophages and monocytes. Within the CCL19 targets, T cells show one of the highest power estimates. Under less optimistic assumptions, the overall power for these gene sets remains below 75%, indicating that larger sample sizes are essential for achieving robust statistical detection. In case an investigation of rare cell types is desired, we recommend sorting with FACS prior to sequencing to enrich these cells enough to have sufficient power to detect differentially expressed genes. Insights like these can streamline resource allocation and inform necessary adjustments to experimental designs.

This framework provides a valuable resource for the research community to plan future experiments more efficiently and cost-effectively. A web-based dashboard (https://scpower.helmholtz-muenchen.de/) allows users to specify their own parameters, e.g., expected cell type abundances, samples, cells or financial budget and others. Additionally, it is possible to obtain power calculations for very specific gene sets or pathways that can be provided based on pilot experiments. This enables researchers with less technical proficiency to first use the web-based scPower tool to plan their experiments and subsequently automatically annotate their data using the web-based archmap tool described above. This user-friendly web-based workflow makes the field of scRNA-seq more accessible to the cardiovascular research community.

## Cell type abundance analysis using single-cell and bulk RNA sequencing

The plaque cell atlas also enables the analysis of cell type abundances across different datasets and sorting strategies. Single-cell data allows researchers to directly assess abundances or deconvolute bulk RNA-seq data for broader insights. For all compositional data analyses, the estimated abundances were centered and log ratio transformed (CLR). The atlas includes datasets from carotid, femoral, and coronary arteries, sorted by different criteria: unsorted, CD45+ cells, or T cells (see Supplementary Fig. 6). This diversity facilitates sanity checks across and comparisons within these groups (see Supplementary Fig. 7). The annotations of the atlas are in line with the expectations, as cells in T cell-sorted datasets are predominantly annotated as T cells and NK cells in our atlas, while CD45+ sorted datasets largely lack structural cells, which are highly abundant in unsorted datasets. Additionally, abundance patterns vary by origin. Monocytes and dendritic cells are abundant in carotid plaques, but nearly absent in the femoral ones (t-test on CLR values: $t = 2.9$, $p = 0.008$, df = 23; $t = -2.5$, $p = 0.02$, df = 23). In unsorted coronary datasets, fibroblasts and ECs are more prevalent, whereas T cells appear less common compared to carotid plaques. A comparison of origins using unsorted datasets is not possible, since the Wirka et al. study is the only one of coronary origin, which confounds the analysis. These findings highlight variability in cell type composition between origins, but should be interpreted cautiously due to potential biases from varying sampling and extraction methods and small numbers of tissue specimens obtained using comparable techniques.

To circumvent this limitation and take advantage of larger sample sizes of bulk RNA-seq experiments, we applied deconvolution of bulk RNA samples using our atlas as a reference. This approach uses cell type specific gene expression information from scRNA-seq as prior to deconvolute bulk samples into a mixture of cell type specific expression profiles and their respective cell type abundances. Moreover, since sample collection and phenotyping is usually harmonized within

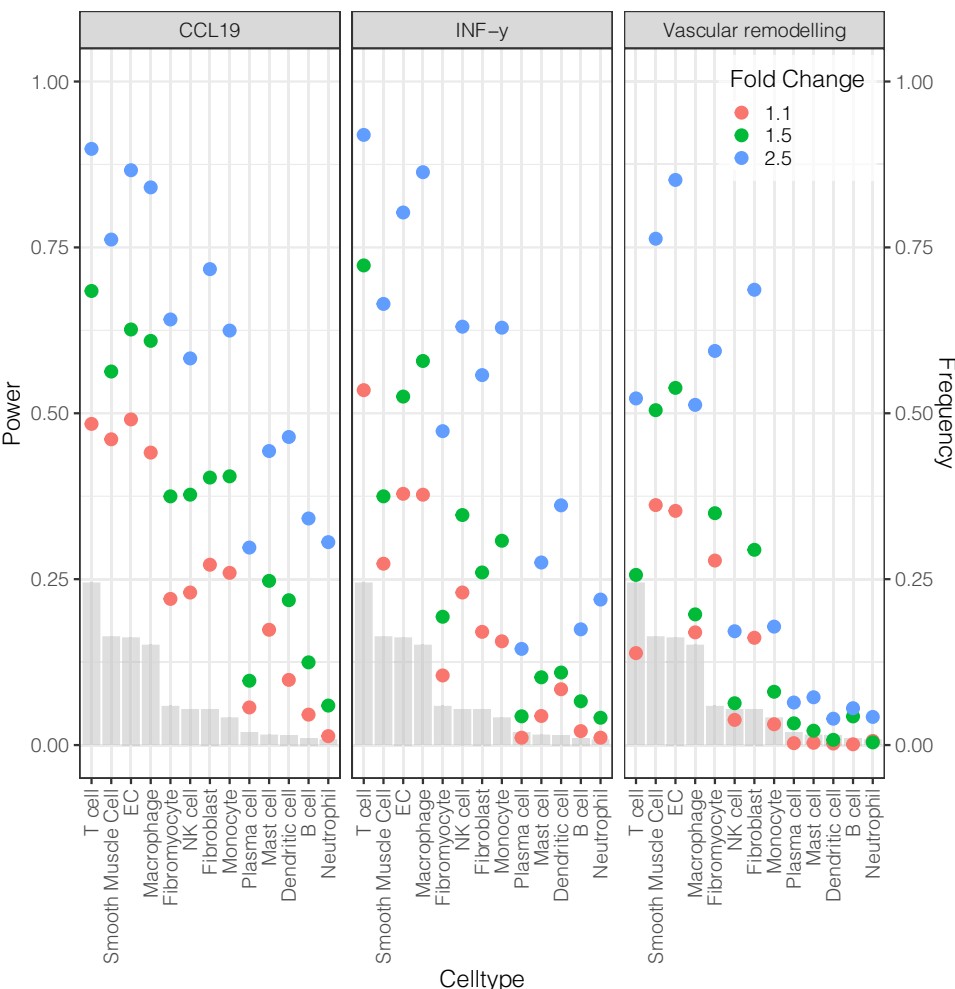

**Fig. 5 | Power analysis utilizing the scPower framework.** The plot shows the statistical power of gene sets related to atherosclerosis with a pessimistic (FC 1.1), neutral (FC 1.5) and an optimistic (FC 2.5) scenario per cell type. The gray bars depict the expected frequency of the cell types in the scRNA-seq study. Source data are provided as a Source Data file.

a study, this approach also eliminates potential confounding due to dataset specific abundance differences. We used BayesPrism[17] to deconvolute a dataset of 236 (202 after QC, see "methods" section for details) bulk RNA seq samples from carotid artery plaque tissue[25].

The plaque samples are classified into early or late (advanced) lesions of stenosed carotid arteries (see "methods"). This enables us to investigate differences in cell type composition between early lesions and diseased carotid arteries. Structural cells, such as fibromyocytes, SMCs, ECs, and fibroblasts together with macrophages, were most abundant (see Fig. 6a). Stratifying by plaque status reveals significantly more SMCs (t-test on CLR values: $p = 2.8e-04$, Supplementary Table 2) in early lesions than in late lesions, and vice versa significantly more fibroblasts (t-test on CLR values: $p = 3.0e-05$, Supplementary Table 2) in late, furthest progressed lesion states. This indicates the high cellular plasticity of SMCs and their transition from SMCs to fibroblast-like cells[4].

We also observe more fibromyocytes in early lesions compared to advanced lesions (t-test on CLR values: $p = 1.9e-03$, Supplementary Table 2), which supports their protective role in atherosclerosis[4].

Additionally, significantly more macrophages (t-test on CLR values: $p = 2.0e-04$, Supplementary Table 2) are observed in late lesions validating the infiltration and increased activity of macrophages in more progressed atherosclerotic lesions and the immune system's contribution to disease acceleration[8,56]. Finally, no significant difference in EC abundance was detectable.

To investigate differences among more granular cell types–referred to as "level 2" annotations–we annotated subclusters of ECs, dendritic cells (DCs), T cells, and macrophages, as these have clearly defined functional subtypes and marker genes (see Supplementary Table 1 and "Methods").

In the EC population, we identified one cluster expressing *ACKR1, AQP1, FABP4*, and *NR2F2*, which can be attributed to a proangiogenic phenotype of venous origin[57,58]. Another EC subcluster expressed established EndoMT genes *COL1A2* and *FN1*, along with co-expression of *GJA4, GJA5, MECOM*, and *GATA2*, indicating their arterial origin[59–61]. Classical EndoMT regulators such as *SNAI1* and *SNAI2* were not expressed in any ECs, except for *SNAI1* in a subset of cells from Wirka et al. (see Supplementary Figs. 8 and 9). This underscores the advantage of simultaneously analyzing multiple integrated datasets and indicates that classical EndoMT regulators are not consistently expressed across all scRNA-seq studies. Because the remaining ECs lacked distinct pro-angiogenic or EndoMT signatures yet expressed *PECAM1* and *VWF*, we labeled them as normal ECs.

We further sub-clustered the macrophages into distinct clusters previously described by a myeloid plaque atlas[8]. Foamy macrophages were classified based on their hallmark gene *TREM2*, as well as *MARCO, FABP4/5* and *CD36*. One cluster was uniquely identified by expression of *TREM1, PLIN2* together with *CCR2* ligand *CCL2* and *CCL7, CCL20, CXCL1/2/3/8*. Another cluster exhibited a mixed inflammatory signature, with *S100A8/S100A9* (calgranulins) and *IL1B* indicating a

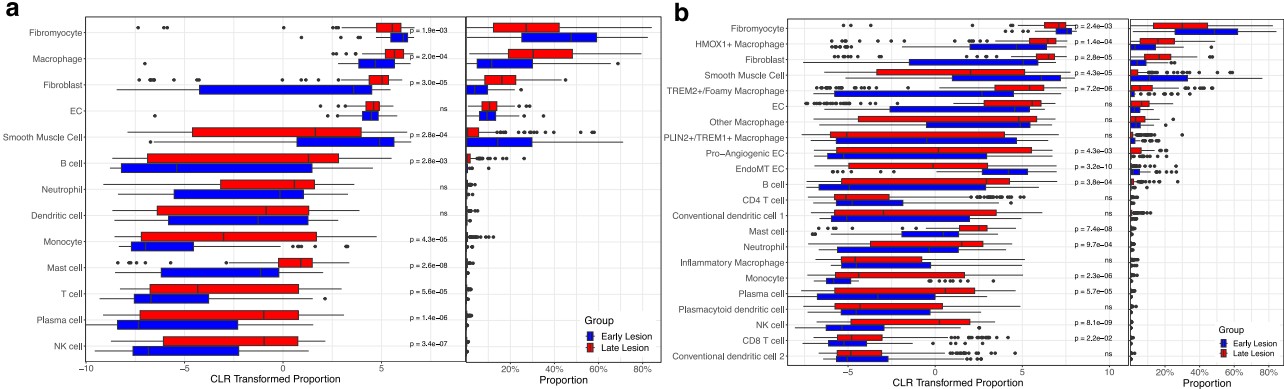

**Fig. 6 | Deconvolution of 202 bulk RNA-seq samples using BayesPrism and the plaque atlas. a** Stratified deconvolution results using the level 1 cell types in the atlas as the reference. **b** Deconvoluted abundances of the bulk samples using the finer grained level 2 cell types in our atlas. (Right) Cell type proportions; (Left) Centered and log-transformed (CLR) proportions to better highlight differences in small abundances. Boxplots depict the median as the center line, box limits at the 25th and 75th percentiles (interquartile range), whiskers extending to the most extreme values within 1.5 × IQR, and individual points beyond the whiskers as outliers. For each of the cell-type comparisons (n = 13 for level 1 and n = 22 in level 2), CLR-transformed abundances in early vs. late lesions were tested by two-sided, unpaired Welch's t-tests (unequal-variance Student's t). Raw p values are displayed, with significance assigned only to those surviving Benjamini–Hochberg FDR adjustment at q < 0.05. Source data are provided as a Source Data file.

strong pro-inflammatory phenotype, while the presence of *IRF7* and interferon-stimulated genes (*IFITM3, ISG15, IFIT2*) points toward engagement of the type I interferon pathway. The absence of chemokines might suggest a less pronounced migratory recruitment function, focusing instead on local inflammatory responses. One cluster displayed a gene expression profile closely resembling the *HMOX1hi* cluster identified by Dib et al. This included genes associated with heme degradation (*HMOX1*), iron processing and export (*FTL, SLC40A1, NUPR1*), and antioxidant activity (*SELENOP, PRDX1*). Additionally, this cluster showed enrichment of lysosomal proteases (*CTSB, CTSD*), lysosomal pathway genes (*LAMP2, LGMN, LIPA, GPNMB*), and genes involved in lipoprotein metabolism (*APOC2, APOE, LRP1, NPC2*). Of note, *HMOX1* was also expressed by the *PLIN2/TREM1* cluster. Lastly, all other clusters that showed none of the above described signatures were termed 'other macrophages' (see Supplementary Figs. 8 and 9).

Dendritic cells formed distinct clusters of conventional dendritic cells 1 (cDC1), expressing *SNX3, CLEC9A*, and *IRF8*, and conventional dendritic cells 2, expressing *CD1C, CLEC10A*, and *FCER1A*. Additionally, there was a cluster expressing plasmacytoid dendritic cell (pDC) markers, including *TCF4, GZMB, TLR7, TLR9, NRP1, SCAMP5, CLEC4C*, and *IRF7*[62] (see Supplementary Figs. 8 and 9).

Finally, T cells were divided into CD4⁺ and CD8⁺ T cells. CD4⁺ T cells prominently displayed *CD4, FTH1, IL7R*, and *ANXA1*, while CD8⁺ T cells highly expressed *CD8A, CCL4L2, CRTAM, CCL4, GZMK*, and *GZMH* (see Supplementary Figs. 8 and 9). Additionally, CD8⁺ T cells showed expression of the CD8 surface protein, which was not expressed in CD4⁺ T cells; conversely, CD4⁺ T cells expressed the CD3 surface protein. This analysis was enabled by the inclusion of the CITE-seq data from Bashore et al. in the atlas.

For the subsequent deconvolution, we used the level 2 cell types with more granular macrophage, DCs, T cells and EC subtypes (see Fig. 6B). The aforementioned differences in SMCs, fibroblasts and fibromyocytes are still visible, while new features are revealed. The newly annotated EC subtypes present with highly significant differences between early and late lesions. While the proangiogenic ECs with venous origin are more abundant (*t*-test on CLR values: *p* = 4.3e-03, Supplementary Table 2) in the more advanced carotid plaques, EndoMT-ECs of arterial origin present the opposing abundance pattern (*t*-test on CLR values: *p* = 3.2e-10, Supplementary Table 2) in early lesion tissues compared to late-stage plaques. One-third (31.6%) of the venous-signature, pro-angiogenic ECs in our single cell reference set originate from carotid plaques. Both the unsorted single-cell carotid profiles and

the bulk RNA-seq samples were obtained by carotid endarterectomy. This procedure excises the atherosclerotic plaque together with adjacent intimal tissue while leaving the outer media and adventitia in situ, thereby ruling out contamination by ECs from neighboring veins in the deconvolution and supporting their derivation from intraplaque neo-vascularization. Together these interesting observations confirm the notion that these EC-rich neovessels in late plaques are allegedly the entry point for immune cells that infiltrate the tissue and likely contribute to lesion progression and instability. This becomes particularly relevant when EC barrier function becomes impaired, neovessels start to leak, and intraplaque hemorrhages occur[63,64].

The stratified macrophage subtype abundance shows that foamy macrophages are, as expected, significantly more present (*t*-test on CLR values: *p* = 7.2e-06, Supplementary Table 2) in the late compared to early lesions, reflecting the important role of macrophage infiltration and activity in late-stage atherosclerotic plaques[8,56]. We did not determine any significant differences in the inflammatory, other macrophages as well as the PLIN2+/TREM1+ cluster. The latter is particularly interesting, as it is associated with vascular events[8], indicating that not the abundance of these macrophages, but rather their gene programs influences disease progression. Another interesting observation is the significantly higher abundance of HMOX1+ macrophages in late lesions (*t*-test on CLR values: *p* = 1.4e-04, Supplementary Table 2), which may reflect the plaque's adaptation to intensified oxidative stress and changing iron-handling associated pathways. In these cells, genes such as *FTL, SLC40A1*, and *NUPR1* are associated with iron metabolism, while antioxidative components including *SELENOP* and *PRDX1* appear to help counterbalance the increased presence of reactive oxygen species. At the same time, the increased expression of lysosomal proteases like *CTSB* and *CTSD*, along with lysosomal machinery genes, such as *LAMP2, LGMN, LIPA*, and *GPNMB*, highlights enhanced proteolytic activity and more robust catabolic processing within these macrophage subtypes. Furthermore, the presence of genes involved in lipoprotein metabolism (*APOC2, APOE, LRP1*, and *NPC2*) suggests an active role in handling and redistributing lipids as the lesion destabilizes. Together, these molecular characteristics imply that as the plaque environment becomes more complex, these specialized macrophages respond by ramping up their iron management, antioxidative defense, and lipid-handling capacities in an attempt to preserve cellular homeostasis.

Overall, bulk to single-cell RNA-seq deconvolution demonstrates the utility of a robustly annotated atlas by highlighting disease-relevant biological processes and comprehensively dissecting cell subtype

abundance with relevance towards atherosclerotic plaque formation and progression.

## Mapping on diverse tissues

Fibromyocytes and macrophages are frequently discussed as relevant cell types for plaque progression[4,65]. Having defined these plaque-specific cell types in our atlas, we next wondered if these cell types are exclusively found in atherosclerotic plaques. Therefore, we investigated 455,953 cells of a total of 23 different scRNA-seq datasets from various organs[24] and assessed their similarity to the cells in the atlas using the automatic mapping tool. Four exemplary organs are showcased in Fig. 7. Interestingly, fibromyocytes were only found in the vasculature in significant numbers (more than 0.22% of total cells), highlighting that cells that were labeled as fibromyocyte are distinct enough to not be detected in other tissues (see Supplementary Fig. 10). Of note, the vascular cells in the Tabula sapiens data set were sampled from individuals with CAD. To investigate the presence of fibromyocytes in non-atherosclerotic vascular tissues, we mapped a dataset of arteries[23] onto our atlas (see Supplementary Fig. 11). The analysis revealed a significant abundance of fibromyocytes in these non-atheroclerotic vessels, indicating that these cells are not exclusive to plaques, but are a normal component of the vasculature, in line with the results of previous studies[27] and our bulk deconvolution analysis in early lesions.

We also screened for foamy macrophages in other tissues and found them highly abundant in accordance with the literature in the lung[66] (see Supplementary Fig. 10). In the original analyses, these cells were mostly annotated as macrophages and their phenotype was not recognized.

As expected, the uncertainty for annotating cells originating from vascular tissues is the lowest, because the atlas includes the surrounding tissues of the plaques as well. In all other organs we observed clusters of cells with very high uncertainty. Further analysis revealed that these clusters are organ specific cells, such as bladder urothelial cells in the bladder, kidney epithelial cells in kidney, and hepatocytes in liver. In the vasculature dataset there was a small cluster of uncertain cells, which were identified as erythrocytes that are not part of our atlas. Simultaneously, the clusters with low uncertainty were correctly assigned by the automatic mapping to the cell type annotated by the authors (see Supplementary Fig. 12). Together this highlights the robustness of our atlas and indicates that atherosclerosis associated cell types can also occur in non-atherosclerotic subjects.

## Discussion

The plaque atlas showcased in this current study represents the most comprehensive curation of single-cell RNA sequencing (scRNA-seq) datasets of atherosclerotic lesions to date[27,28]. This atlas, extensively validated through annotation consensus, protein measurements, and the illustration of known biological processes, serves as a robust reference for future studies. It provides a foundation for integrating new research questions and advancing our understanding of atherosclerosis in the hunt for novel biomarkers and therapies.

Previous integration efforts for single-cell data from plaque tissue[27,28] were often constrained by limited scope, validation, and usability. In contrast, our atlas encompasses all major lesion locations, offers thorough orthogonal validation, and enables user-friendly exploration via an interactive web interface. By integrating diverse datasets, we were able to identify neutrophils, which were overlooked in most of the individual datasets, as it is notoriously difficult to identify in scRNA-seq data[45,46]. Interestingly, our neutrophil cluster consisted almost exclusively of cells from the Slysz et al. dataset[9] which is CD45+ sorted. This highlights differences in sensitivity among various datasets and protocols used and underscores the advantage of integrating them into an unified atlas. Here we propose new human plaque marker genes to robustly identify this cell type in all future

studies. Despite having defined granular level 2 cell type annotations for the deconvolution, we recognize that the consensus on these annotations can vary depending on the specific research question. Often, these specific subtypes are transcriptionally highly similar, which makes it difficult to unambiguously distinguish them. Hence, we recommend using level 1 annotations initially, followed by subclustering into level 2 cell types of interest, as demonstrated in our bulk deconvolution workflow, which yielded results in line with previously published studies from the cancer field[17].

The scPoli model, integral to this atlas, is trained on predefined cell types, limiting its predictive capacity to those included in the study. The level 1 cell type annotations are in high concordance with the author-provided annotations and represent the consensus in the field. Thus, we expect this level of annotation to remain stable. Level 2 cell type annotations are currently limited to those subtypes that have well established functional roles and established markers which clearly separate cell type clusters in our atlas. These level 2 annotations heavily depend on experimental evidence and are likely to evolve as new findings emerge. Consequently, additional experimental evidence will also uncover new functional roles for specific cell subsets. In case of a new single-cell study that includes a novel sufficiently distinct subset of cells, the model provides uncertainty metrics to flag these cells. Clusters of cells with high model uncertainty could indicate the presence of yet unidentified cell types not included in the atlas, as demonstrated with the non-plaque tissue mappings.

Overall, this atlas not only serves as a robust foundation for future research but also enhances accessibility for newcomers to the field, by making it easy to use. Our atlas can be integrated into the training of foundation models[20–22], thereby expanding the dataset corpus to include plaque tissues. This inclusion is crucial, as current models often underperform in out-of-distribution tasks, reflecting an overly homogeneous training corpus.

We demonstrated four primary downstream tasks: automated cell type annotation, power analysis and study planning, cross-organ analysis and abundance analysis with single-cell data and bulk RNA sample deconvolution. For future studies, the atlas can be used in additional downstream applications, such as integrating scRNA-seq data with spatial transcriptomics datasets and conducting cell type specific genome-wide association studies using prior atlas information[27]. Given the availability of specially designed bulk RNA-seq experiments, the atlas can facilitate the deconvolution of phenotype-specific signals or survival analysis. Reference mapping, including healthy reference atlases, can highlight differences between healthy and diseased tissue samples at the single-cell level, as recently emphasized[67]. This underscores the necessity of an atlas comprising exclusively diseased samples.

We acknowledge that there are certain limitations. First, the representation of cells of different arterial origins is not balanced. In particular, femoral arteries were only sampled in a single study, where a pre-selection for CD45 positive cells was applied. Thus, structural cells in femoral arteries could not be sufficiently characterized. In general this scarcity of cells of femoral origin hinders comprehensive comparisons across different vascular beds and origins. Second, the lack of a clearly defined dataset of "healthy" arteries makes comparisons between healthy and aberrant cell states difficult. Third, the quality of cell type annotations depends on the publicly available marker genes. While we mitigated this issue by incorporating a consensus annotation from independent experts and surface marker measurements, the experts likely relied on similar marker genes. Last, the deconvolution approach is limited to cell types present in the reference atlas. While we believe all major cell types described in the literature are covered, newly discovered cell types first need to be annotated in the atlas before cell type proportions can be deconvoluted. Both bulk and scRNAseq can have technical biases that can skew the observed proportions of cell types within tissues. However,

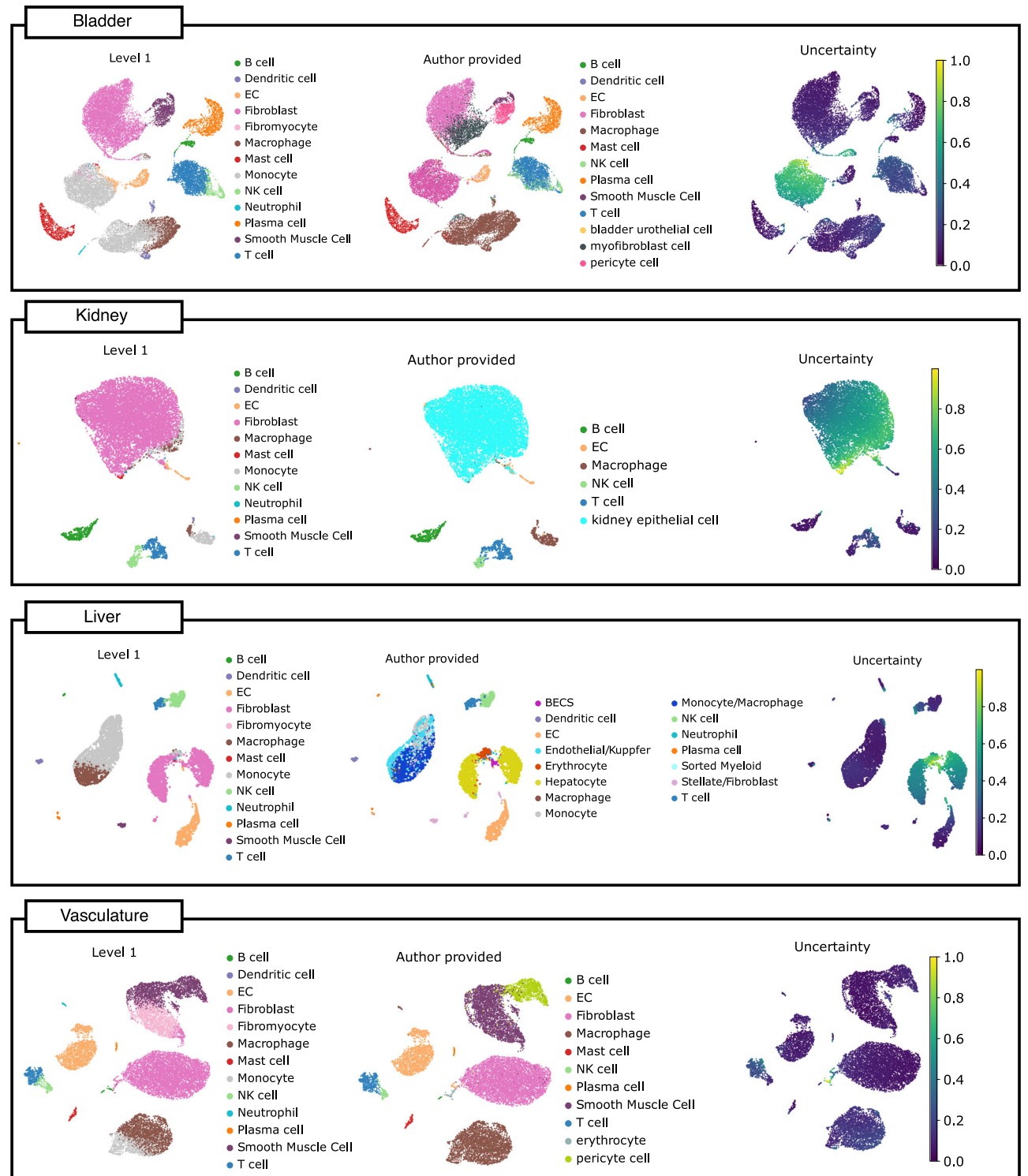

**Fig. 7 | Mapping Tabula Sapiens organs scRNA-seq datasets on plaque atlas.** Four exemplary mappings of the Tabula Sapiens dataset on the plaque atlas. For each organ, the predicted level 1 cell type is shown on the left panel, the models uncertainty on the right and the free annotations provided by the Tabula Sapiens authors in the middle.

because samples from the same technology share the same bias, comparisons within technologies still reveal meaningful relative differences while comparisons across technologies have to be treated with caution and are not advised. In cases where detailed characterization of structural cell populations is required, single-nucleus RNA sequencing may be an alternative to scRNA-seq[68].

In summary, this comprehensive and robust atlas serves as a powerful resource, unlocking countless opportunities for downstream applications and enabling the discovery of novel cellular processes in atherosclerosis. Importantly, it highlights that fibromyocytes are specific for vascular tissues, indicates expansion of pro-angiogenic endothelial subsets, and shows enrichment of HMOX1+ macrophages

in advanced plaques. These findings reveal potential cellular mechanisms of lesion progression and instability that warrant follow-up investigations. Looking ahead, the atlas will be regularly updated to incorporate new findings, ensuring it remains at the forefront of advancing our understanding of this complex disease.

## Methods

### Preprocessing of all data sets

In all datasets QC was applied on the uncorrected counts. Doublets were tagged with scDblFinder[30] with the sample as the batch parameter and the counts are corrected for ambient RNA with celda[69] using the sample as the batch parameter and rounded to integers. The uncorrected counts and celda corrected counts were kept as separate layers.

### Preprocessing of Pan et al.[70]

The three samples were downloaded from GEO and came already filtered with 200 <#genes <4000, maximum counts of 20.000 per cell and mitochondrial gene count percentage lower than 10%. The three samples were outer joined with the *concat* method of the anndata[71] package.

### Preprocessing of Alsaigh et al.[26]

The dataset was downloaded from GEO and only plaque samples (barcode suffixes 2, 4 and 6) were taken and adjacent tissue samples excluded. The samples were renamed from 2, 4, and 6 to 1, 2, and 3 respectively. Cells were filtered on the uncorrected counts according to the original publication with 200 <#genes <4000 and percentage_counts_mitochondrial >10%. Additionally, we filtered out genes expressed in less than 3 cells.

### Preprocessing of Fernandez et al.[5]

The dataset was downloaded from GEO and each sample is read in with scanpy[72] and outer joined with the concat method from anndata. As sample 6 is a CITE-seq sample it also includes surface protein markers. These were removed from the dataset as the atlas is only based on gene expression. We applied our own filtering on the uncorrected counts and filtered out cells with less than 200 expressed genes, 500 <#counts <40000 and more than 10% mitochondrial gene percentage. Genes were filtered out that are expressed in less than 3 cells.

### Preprocessing of Pauli et al.[25]

The dataset was preprocessed with Seurat with nFeature_RNA > 200 & nFeature_RNA < 10000 & nCount_RNA > 1000 & nCount_RNA < 50000 & percent.mt <20. Only the diseased samples were taken and the adjacent samples excluded.

### Preprocessing of Dib et al.[8]

The dataset was downloaded from GEO and the sample ids extracted from the barcode suffixes. The sample ids were changed from 5 to 4, sample 6 to 5 and sample 7 to 6, but kept sample IDs 1, 2, and 3 the same. Cells were filtered on the uncorrected counts with more than 400 genes expressed, 1000 <#counts <30000, and pct_counts_mt <10%. Genes that are expressed in less than 3 cells were filtered out.

### Preprocessing of Slysz et al.[9]

The samples were downloaded from GEO. Femoral and carotid samples were loaded in and concatenated with *concat* of anndata into a femoral and carotid anndata object. For both objects the cells were filtered on the uncorrected counts according to the authors with 200 <#genes <10000, 200 <#counts <10000 and pct_counts_mt <10.

### Preprocessing of Ahmad et al.[48]

The dataset was downloaded from GEO. We selected the "Fresh_ROB_2026" and "Fresh_DTAN_4047" samples and concatenate them into one dataset with *concat* from the anndata package. Metadata was

also available and cells which had "Removed by QC" as cell types were removed. To be as consistent as possible we converted the provided ensembl gene ids in this dataset into gene names using the mapping created out of the Fernandez et al. dataset[5]. 4250 genes found no mapping and were removed and kept 32251 genes. Cells were additionally filtered on the uncorrected counts by us with min_genes = 200, 500 <#counts <40000 and pct_counts_mt <10, while genes that are expressed in less than 3 cells were excluded as well.

### Preprocessing of Emoto et al.[6]

The dataset was downloaded from GEO and the SAP and ACS samples were preprocessed according to the author with min_genes = 500, min_cells = 3, pct_counts_mt < 8 and max_genes = 5000. Subsequently, both datasets were concatenated with concat from the anndata package. The ACS samples are termed sample 1, while the SAP samples are termed sample 2.

### Preprocessing of Chowdhury et al.[7]

The dataset was downloaded from GEO and came already filtered with the following criteria: Genes were filtered that are expressed in less than 5 cells. Cells were filtered with more than 38% pct_counts_mt. Cells for 10x v2 samples were filtered with min_genes = 300 and max_counts = 15.000, while 10x v3 samples were filtered with min_genes = 500 and max_counts = 20.000. The blood samples were excluded and only the plaque samples were kept. The sample ids were changed from alphabetical to numerical according to the letters position in the alphabet.

### Preprocessing of Wirka et al.[4]

The dataset was downloaded from GEO and came already filtered with the following criteria: Genes expressed in less than 5 cells were filtered out. Cells were filtered with pct_counts_mt <7.5% and kept with 500 <#genes <3500. There were duplicate barcodes which we investigated more closely. They had different gene expressions, hence we assumed they are different cells and made the barcodes unique.

### Preprocessing of Bashore et al.[47]

The dataset was downloaded from GEO and only the gene expression data from the samples was read in and concatenated with *concat* of the anndata package. It was filtered according to the authors with 200 <#min_genes < 6000, max_counts = 40.000, min_cells = 3 and pct_counts_mt < 30. For this dataset the doublet tagging ambientRNA correction was applied shortly before the reference mapping.

### Manual annotation of samples

Because samples within datasets can also entail batch effects due to the data collection or other factors, we annotated and integrated on a sample level. We manually annotated sample 1, 2, and 3 of Alsaigh et al., sample 2 and 3 of Pan et al., sample 6 and 8 of the Slysz et al. femoral dataset and sample 5, 6, 7, and 8 of the Wirka et al. dataset. For all datasets we performed *scran*[73] normalization with an initial clustering of total counts normalization with 1e6 as target sum, log1p normalization, PCA, neighborhood calculation on 30 PCs and Leiden clustering with a resolution of 0.22. The resulting size factors were used to normalize the counts. Subsequently log1p transformation is applied, top 2000 highly variable genes selected, PCA and neighborhood calculation on 30 PCs. Then UMAPs are calculated for each sample and clusters were manually annotated with the level 1 marker genes. Doublet clusters were annotated as doublets. Clusters with no distinct marker gene patterns were annotated as unknown. In cases of uncertainties of cluster annotations, these clusters were sub clustered and annotated on a finer resolution.

### Benchmark of integration methods

All annotated samples are concatenated into one dataset and the gene names are mapped to Ensembl ids to solve the problem with changing

gene names and aliases. Duplicate genes after the mapping were aggregated. The resulting dataset was normalized with *scran* using the same parameters as in the manual annotations and log1p transformed. The sample ids were suffixed with the dataset name to make the sample ids unique. Unknown and doublet cells were removed. It was preprocessed with the reduce_data method with sample id as batch_key of the scib[31] package. Highly variable genes were selected and cells with zero counts were removed. Additionally, cells with the duplicate gene expressions were removed as well. The methods that are benchmarked are scVI[32], Harmony[33], LIGER[34], scANVI[35], scGen[36], scPoli[37] and baseline PCA. For scGen, the scib implementation was used on the log-normalized counts with the sample id as the batch key and the manual annotations as cell types. For scVI the default parameters and counts are used. Subsequently the scVI model is fine-tuned with scANVI with the cell type labels. For scPoli, the default parameters are used, but the loss is changed to "mse" and calculated on the log-normalized counts. Harmony is calculated on the PCA embedding with the harmony-pytorch package. LIGER embeddings are calculated following the tutorial on scib-metrics[31]. The resulting embeddings are benchmarked with the scib-metrics benchmarker using the sample id as batch key and cell types as label key.

## Atlas integration

To generate the whole atlas all preprocessed datasets except Bashore et al. were concatenated with *concat* from anndata. The sample ids were made unique by suffixing the dataset. Subsequently the gene names were harmonized by mapping them to ensembl genes and aggregating duplicates. Duplicate and zero count cells were removed and the whole dataset was *scran* normalized and log1p transformed with the parameters described earlier. Manual annotations were transferred from the benchmark subset. For the preprocessing of the whole object, the same pipeline as in the benchmark was applied. A split of reference and query dataset was performed where the manual annotations serve as the reference. A scPoli model was trained on the reference dataset log-normalized counts with the sample as the condition key and the cell type annotations as cell type key. An embedding dimension of 10 was chosen and trained with a mse reconstruction loss. For everything else default parameters were used. Subsequently, the query dataset was mapped to the reference using the pretrained reference scPoli model and did transfer learning with scArches and new labels including uncertainties predicted. Afterwards, all level 1 cell type clusters were reanalyzed independently to validate the robustness of the annotations. This resulted in the identification of neutrophils, which was overlooked in the previous annotations. Unsure cells were labeled as unknown and subsequently remapped to the reference.

## Validation of annotations (expert and CITE-seq)

To validate the annotations, an expert annotated three datasets (Pauli[25], Emoto[6] and Wirka[4]) independently from us without our help or interaction using his own methods and judgment. One dataset[48] was annotated by the authors of the original publication. To compare the cell type annotations with our predictions, they needed to be harmonized. For this, we used our level 1 cell types and did the same to the provided annotations from the expert and authors. The intersection of cell types between our predictions and the author/expert provided annotations were used to calculate confusion matrices and precision/recall. The precision and recall was calculated per cell type and then weighted according to their abundance to yield an overall precision and recall per dataset. For the CITE-seq dataset we looked at the protein measurements and used the *dotplot* function in scanpy to plot our predicted cell types in these samples to the surface markers that we grouped into cell types, while scaling the expression between 0 and 1 within each surface protein (default).

## Automatic cell type annotation on Bashore dataset

The gene expression data of the Bashore et al.[47] dataset was used to validate the automatic cell type annotations. We made the sample names unique, mapped the gene ids to ensembl ids with our mapping, removed 22 non-mapped genes and aggregated duplicated genes (same workflow as before). Our standard *scran*-log1p normalization/transformation was applied. The dataset was then subsetted for the 2000 genes used in the scPoli model, where 9 genes were not in the query dataset. The missing information was filled with zeroes. This resulted in 77.112 cells that were loaded into the reference model and trained with scArches. Finally, cells with uncertainty higher than 0.7 are removed. The same pipeline for validation with confusion matrices and protein surface markers as in the previous section was applied, where the author provided us with the cell type labels. Cells that were not given a label by the authors were labeled as "unknown" by us. The atlas including the Bashore dataset was used for subsequent downstream tasks.

## Power analysis with scPower

To calculate the gene expression priors the uncorrected counts and level 1 cell type annotations were used. The gamma and dispersion fits were calculated per cell type. The matrices were subsampled using multinomial sampling to simulate 25%, 50%, and 75% total counts. Because the T cell matrix was too big in memory, we used a more memory efficient subsampling method that uses sparse matrices instead of dense matrices. Subsequently, the gammas and dispersion priors were fitted using the tutorial in the vignette of scPower. To model the transcriptome mapped reads to UMI relationship we used the Pauli et al. dataset as priors where we had CellRanger outputs. The same holds for the cell type frequency priors, for which we used the predictions of the atlas in this dataset. We assumed 22 samples with 200 cells each, a read depth of 334.015 and mapping efficiency of 43% which were the mean parameters in our samples. All of these parameters can be easily changed in our interactive web-based dashboard according to your experimental setup. The three gene sets used were: (1) targets of the CCL19 chemokine and its targets downstream of CCR7. To get these we downloaded the ligand-target matrix from https://zenodo.org/record/3260758/files/ligand_target_matrix.rds and extracted the top 50 genes of the CCR7 targets with the highest score; (2) genes of the Interferon-Gamma (IFN-y) pathway which where downloaded from https://www.gsea-msigdb.org/gsea/msigdb/human/geneset/HALLMARK_INTERFERON_GAMMA_RESPONSE.html; and (3) vascular remodeling genes, which were identified through genome-wide association studies of coronary artery disease in Chen et al.[55].

## Sample collection bulk RNAseq data

Sample collection and preparation for sequencing was performed as described before[25]. The classification of human carotid artery samples into early and late-stage atherosclerosis was initially performed visually during the cutting and preparation process. This means the identification and separation of advanced plaque parts with small lumen versus large lumen without visually detectable plaque. The classifications were then confirmed through subsequent histomorphological analysis of FFPE-processed adjacent sections. Lesions were categorized according to AHA guidelines[74–77] as previously described[25]. Samples classified as I, II and III were designated as early lesions, while those classified as V to VIII were assigned late lesions.

## Preprocessing bulk RNAseq data

We reprocessed the data from Pauli et al.[78] and performed adaptor clipping and quality trimming of sequences using Fastp v0.23.2 (https://github.com/OpenGene/fastp). Subsequently, reads were aligned to the GENCODE v40 GRCh38 reference transcriptome using the transcript quantifier Salmon v1.6.0 (https://salmon.readthedocs.io/). The R package tximeta was employed to combine Salmon transcript quantifications with sample data and to summarize transcript quantifications at the gene level.

To assess data quality at both the read and alignment levels, we utilized FastQC v0.11.9 (www.bioinformatics.babraham.ac.uk/projects/fastqc/) and MultiQC v1.13a (https://multiqc.info/) to generate statistics before and after trimming. Outlier samples were detected by performing a principal component analysis (PCA) on variance-stabilizing transformed counts for the 500 genes with the highest variance, using the R package DESeq2. The resulting counts can be found in Supplementary Data 1.

Based on the QC metrics, we removed samples that met the following criteria: PCA outliers (determined by visual inspection), percentage of mapped reads <75%, percentage of duplication > 60%, and number of mapped reads <10,000,000. After filtering, a total of 202 samples remained.

### Cell type abundance analysis using single-cell and bulk RNA sequencing

For finer grained level 2 cell types we used the level 1 predictions from the atlas integration. We selected all cells of a certain cell type, took the rounded corrected counts layer and applied our standard preprocessing pipeline to log-normalize it. Harmony was selected over other integration tools due to its effectiveness in reducing batch effects without relying on predefined cell types. This allowed for an unbiased identification of sub-cell types by minimizing the risk of clustering biases within Level 1 categories. Harmony provided an integration with good biological signal retention, which we used solely for reannotation; the integrated embedding was then discarded, and only the updated cell annotations were used in further analyses. The resulting dataset was clustered with the Leiden algorithm and manually assigned to cell types with our level 2 marker genes. This was done for macrophages, dendritic cells, T cells and ECs. We were also looking for potential mis-classifications and corrected them by assigning the correct cell type. Cells with no clear signature ($n = 264$) were labeled as "Undefined" and removed from the atlas in downstream tasks.

To deconvolute the bulk samples we used BayesPrism and closely followed their tutorial provided in the GitHub repository (https://github.com/Danko-Lab/BayesPrism/). The uncorrected counts of the atlas including the cell type annotations were loaded and Ensembl IDs were used in both the scRNAseq reference and the bulk matrix. Genes were first filtered with the cleanup.genes function with default parameters, then filtered for protein coding genes with select.gene.type and finally signature genes were calculated with get.exp.stat and subsequently filtered with select.marker. For the level 1 deconvolution the level 1 cell types were used as the cell.type.labels and the level 2 cell types used as the cell.state.labels. For the level 2 deconvolution both the cell.type.labels and the cell.state.labels were set to the level 2 cell types in the atlas. Finally, the theta values for each cell type per sample are used for the figures. The cell types are sorted according to their mean proportions across all samples. The abundances were centered and log transformed with the clr function of the compositions[79] package. For significance testing a $t$-test with default parameters was applied on the CLR transformed values.

### Mapping on diverse tissues

We retrained the scPoli model separately with the level 1 and level 2 annotation as the reference. All datasets from the Tabula Sapiens dataset were downloaded from Zenodo. All tissues were put into our automatic cell type annotation scripts provided in the GitHub repo, which includes normalization, selection of genes, optional renaming of genes to Ensembl IDs, and finally the mapping to the atlas. For abundance analysis cells with a higher uncertainty then 0.5 were removed to exclude uncertain cells, while for the confusion matrices we opted to use an uncertainty threshold of 0.7 to include more uncertain cells. To make the comparisons of the author provided labels with our predictions, we renamed the author provided labels to match our ontology.

### Reporting summary

Further information on research design is available in the Nature Portfolio Reporting Summary linked to this article.

### Data availability

The scRNAseq data used in this study is all publicly available under the provided references in Table 1. The fully processed atlas is available through the CELLxGENE portal at https://cellxgene.cziscience.com/collections/db70986c-7d91-49fe-a399-a4730be394ac. The processed bulk dataset including metadata used in this study is accessible in Supplementary Data 1. Source data are provided with this paper.

### Code availability

The Python and R code to reproduce the results is available at https://github.com/heiniglab/reproducibility-plaque-atlas (https://doi.org/10.5281/zenodo.15389565). The Python script and Docker container for the automatic cell type annotation are available at https://github.com/heiniglab/plaque-atlas-mapping (https://doi.org/10.5281/zenodo.15389569)[80] and https://github.com/matmu/plaque-atlas-mapping_docker. The WebUI is accessible at https://www.archmap.bio/#/genemapper/create. To use the tool, select "Plaque" as the atlas and "scPoli" as the model. The platform will then provide detailed instructions for uploading and preparing data for annotation.

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

## Acknowledgements

K.T. is supported by the Helmholtz Association under the joint research school "Munich School for Data Science—MUDS". L.M. is supported by the ERC Consolidator Grant LongTX (under the grant agreement number 101088370), the Bavarian State Ministry of Health and Care through the DigiMed Bayern project on P4 medicine, and the DFG-funded TRRs/CRCs 267 (Non-coding RNAs in the cardiovascular system) and 1123 (Novel targets in atherosclerosis). M.H. is supported by the Chan Zuckerberg Foundation (2019-202666, 2021-237882) and the DZHK (German Center for Cardiovascular Research) projects 81Z0600106 and 81Z0600105. We would like to thank Malte Lücken for insightful discussions on methodological aspects of building single-cell atlases.

## Author contributions

Conceptualization: M.H., K.T., P.K.; Formal analysis: K.T., M.M.; Writing—Original Draft Preparation: K.T., M.H., J.P., L.M., M.M.; Writing—Review and Editing: all authors; Supervision: T.C.R., L.M., P.K., M.H.; Resources: L.M., N.S.; Data Curation: K.T., E.V., J.P.; Funding Acquisition: P.K., M.M., T.C.R., M.H., L.M.

## Funding

## Competing interests

M.M., E.V., T.C.-R. and P.K. are employees of Roche Diagnostics GmbH, Penzberg, Germany. The remaining authors declare no competing interests.
