## [Transparent Peer Review file · Nature Communications]

Integrated single-cell atlas of human atherosclerotic plaques

Corresponding Author: Dr Matthias Heinig

Version 0:

Reviewer comments:

Reviewer #1

(Remarks to the Author)

Traeuble et al present an integrated single-cell atlas of human atherosclerotic plaques comprised of 261,747 cells from carotid, coronary, and femoral arteries. The authors manually annotated a subset of samples, applied the best-performing integration method to build the atlas, and subsequently subclustered 'level 1' cell type annotations to identify more granular cell subsets (termed 'level 2' annotations). Finally, downstream applications of the constructed atlas are demonstrated, including automated cell type annotations and bulk RNA sample deconvolution.

While this study has the potential to serve as an informative resource, the manuscript in its current format reads like a methods paper rather than offering unique biological insights into the cellular plasticity driving advanced plaque progression. A more thorough analysis of the normal versus aberrant cell states underpinning atherosclerosis would be expected from a designated atlas paper. Specific comments and suggestions of this reviewer are provided below.

Major Comments:

Atlas datasets

1. The authors state that current single-cell atherosclerosis atlases are limited to only carotid and coronary artery samples. However, only one femoral artery dataset is included in this atlas (GSE234077). Moreover, based on the experimental design of this dataset, single-cell suspensions of the plaques were enriched for CD45+ leukocytes for subsequent sequencing. In this regard, this dataset may not be sufficient to provide comprehensive coverage of all cell types present in plaques from femoral arteries.

2. This atlas excludes healthy samples. Can the authors further comment on this decision? Including both healthy and diseased samples in the reference atlas would allow for detection and comparison of healthy vs disease-specific cell types.
a. A random selection of healthy cells (to a pre-defined number) can be implemented to avoid over-representation of healthy cells in the integrated data.

3. Do the included single-cell atlas datasets include metadata about the severity of the disease (e.g. early vs late lesions)? If so, the manuscript would benefit from the added biological insights throughout disease progression. Further, the single-cell data could validate the results of the bulk RNA deconvolution between early and late lesions.

Cell type annotation

1. How was the level of uncertainty quantified for cell type assignment?

2. Some of the marker choices for cell type annotation require additional explanation and/or review.

a. Pro-angiogenic ECs:

Why are proliferation markers or collagen/remodeling markers not included for this annotation?

Please provide references that the provided level 2 pro-angiogenic markers are indeed canonical for this subset.

b. EndoMT ECs:

The authors utilized markers characteristic of mesenchymal cells. EndoMT regulators such as Snail1 and Snail2 should also be considered.

3. A goal of the study is to enable accurate automatic cell type annotation of new datasets. However, the atlas does not correctly identify neutrophils, which are reported as known cell players in atherosclerosis. Can the authors comment on this limitation? Could more stringent cell annotations and/or the inclusion of additional datasets be considered to ensure these

subsets are not overlooked?

Methodology

1. The authors identify scPoli to be the best-performing integration method in bio conservation and batch correction. Why then is Harmony used to integrate samples for the level 2 annotations (only briefly mentioned in 'Level 2 Annotation' subheading of Methods section)?
2. An integration pipeline starting with manual annotation of a subset of samples followed by scPoli integration and label transfer is employed. Is such a sequence (annotation first, integration after) a standard in single-cell analysis comprising multiple datasets that require batch correction?
3. While the scPower framework may serve as a useful resource for planning future experiments, this section in the text comes across quite randomly in the paper's current format. Moreover, it is not entirely relevant to understanding the complexities underlying atherosclerosis in this manuscript.

Biological Insights/Questions

1. Are there different cell types (abundances and/or altered phenotypes of a specific subset) present between carotid, coronary, and femoral samples? There is currently no information provided from the single cell studies to understand these origin-specific plaque differences at the cellular level.
2. In Supplemental Figure 3, the clustering of lymphocytes in the UMAP (NK, T cells, and B cells) overlaps with the cells from femoral artery samples more so than within the myeloid populations. Is this in line with previous studies?
3. Can the authors comment further on the finding of venous ECs in the advanced plaques? Specifically, does the mapping of pro-angiogenic ECs to venous EC origin definitively indicate neovascularization within the plaque? Or is it possible that surrounding veins were collected in addition to the analyzed arterial sample?
4. Atlas papers typically require a detailed description/proposed function of each cell type comprising the atlas. A much more in-depth analysis regarding the annotated cell types and their role in disease progression is expected. If this is not the goal of the manuscript, "atlas" should be removed from the title.

Minor Comments

1. A scale should be provided for the dot plots in Figure 2 and 4 (as done for Figure 3 E).
2. Figure 2: CD8A is listed twice in the level 2 annotation criteria for CD8 T cells.
3. Page 6, 1st paragraph: "The predicted cell types in our atlas were compared to the provided labels. The precision of the annotations is 89.37% in the Pauli dataset... (see Figure 4 A-D)." This should be corrected to Figure 3. Same comment when referencing panel E later in the same paragraph.
4. Page 20: "...inflammatory and foamy macrophages are substantially enriched (P-value)..." Provide the p-value.
5. Higher resolution / better quality needed for Figure 6.

(Remarks on code availability)

Reviewer #2

(Remarks to the Author)

In this study, the author presents an integrated single-cell data source of 261,747 cells from human atherosclerotic plaques. Robust cell-type annotations of cells in the data source are provided and further validated independently. In the applications, the author demonstrates that the atlas enables accurate automatic cell type annotation of new datasets, optimal experimental design, and deconvolution of bulk RNA-seq data to determine cell type proportions in human atherosclerotic lesions. However, the following concerns should be addressed before publication.

1. It is great that the author provides a comprehensive scRNA-seq data source for the study of human atherosclerotic carotid, coronary, and femoral arteries. However, a crucial concern is whether the data is comprehensive enough to contain all possible cell types or subclusters for the three types of arteries. If not, how does this issue affect the downstream analysis (i.e., cell type annotation and deconvolution)? For instance, if there are novel cell types or subclusters in new datasets, which can happen in practice, can we still use the atlas to perform cell type annotation? Similarly, can the atlas still be used for cell type deconvolution when bulk RNA-Seq data contains the cell types that are novel to the atlas?
2. A key benefit of the data source is to provide accurate cell type annotation, which is also the basis for cell type deconvolution with bulk RNA-seq data. A crucial concern is whether the data source can provide a superior performance in cell type annotation of new datasets, compared to the computational tools (e.g., scVI, Harmony, scPoli, scGen, scANVI, etc.) for cell type annotation, which are mentioned in the manuscript. Otherwise, there is no reason for researchers to resort to the

data source for cell type annotation.

3. On page 5, the author provides a curated table of plaque-specific marker genes for both level 1 and 2 cell types. How do the authors obtain the marker genes? The list can be incomplete. If incomplete, how are the results affected?

Moreover, as stated in the manuscript, all 11 samples are manually annotated, but as shown in the suppl. figure 2, the comparison was conducted on a subset of manually annotated samples. I think the scalability of these methods is not a concern, and the author does not provide us with details on the selection of samples. Why was the comparison conducted on a subset of samples? How was the subset of samples selected?

4. In the level 2 annotation, Harmony was used to eliminate potential batch effects and conserve as much biological signal as possible. However, in the benchmark of different integration methods, compared with other methods, Harmony does not perform better in bio conservation and batch correction, as shown in Suppl. Figure 2. Why did the authors use Harmony, instead of other methods (e.g., scGen and scANVI) for the purpose?

5. In the manuscript, the authors provide an easy-to-use interactive Web UI to annotate new datasets including uncertainty automatically. However, the website for the tool is not provided in the code availability section. It should also be accompanied with a detailed documentation.

Some minors:

1. There are several typos in number formatting, especially in Table 1. For example, 8.867 in Table 1 should be 8,867. Please check the manuscript to correct similar typos.

2. Some citations are missing. For example, there is no reference in the sentence “Subsequently, we looked for finer-grained cell types, which we refer to as “level 2”, where we focus on cell types that have distinct biological functions that have previously been described in the literature.” on page 15.

3. For producibility and reuse, some documentation for the GitHub repository (<https://github.com/kotr98/reproducibility-plaque-atlas>) should be provided.

(Remarks on code availability)

It seems to me that the code is reasonably detailed for reproducibility.

Version 1:

Reviewer comments:

Reviewer #2

(Remarks to the Author)

All my previous comments have been reasonably addressed. I do not have any further comments.

(Remarks on code availability)

Reviewer #3

(Remarks to the Author)

In this study, Traeuble and coworkers undertake an integration of 259,493 cells from human atherosclerosis plaques from a range of different studies. They provide a tool for the annotation of cell types for the processing of single cell data from human plaques.

As I am taking over the role of Reviewer #1, I will solely comment on the concerns raised by Reviewer 1.

While the authors have addressed many of the concerns raised by Reviewer 1, there are a couple of critical points that – in my view – the authors have not addressed.

- Reviewer 1 raised in at least 3 different points the lack of “unique biological insights into the cellular plasticity driving advanced plaque progression”. While I appreciate the extended discussion of cell types from deconvoluted RNA-seq data, it is unclear to me whether this study provides any novel insights into mechanisms driving plaque progression.

- Reviewer 1 raised the concern of accurate cell type annotations multiple times. I still hold this concern for the endothelial (EC) populations, where the authors have compartmentalised all ECs into “proangiogenic ECs of venous origin” and “EndoMT ECs”. I find it strange that no “normal” aortic ECs are annotated despite the atherosclerotic tissue. As EC dysfunction is a key driver of atherosclerosis development, further subclustering of EC clusters is required to better define this population.

Along these lines, the authors find that “fibromyocytes” make up ~50% of cells in the plaque in their deconvolution analyses

(Fig. 6A,B). Is this expected? It seems contradictory to their UMAPs of integrated data from the various scRNA-seq studies. Is this due to inaccurate markers for this population or methodological issues with the deconvolution?

- Reviewer point 1: The authors repeatedly point out the lack of scRNA-seq data of femoral plaques in their response. However, they have also overlooked PMID: 36547462, which analyses ~14k cells from femoral plaques. Is there a reason that the authors do not include this study, which will allow better comparisons across plaques?

Minor points

- Methodology- reviewer point 3: while I have not read the original manuscript, I share this concern. The scPower section does come out of the blue and it is not clear how it adds to the overall story presented by the authors.

(Remarks on code availability)

Version 2:

Reviewer comments:

Reviewer #3

(Remarks to the Author)

The authors have addressed all my comments.

One small recommendations given the high number of fibromyocytes in the bulk versus scRNA-seq data - the authors could mention in their discussion that it may be more advantageous to evaluate atherosclerotic plaques via single Nuclei RNA-seq rather than scRNA-seq. This may preserve all cell types better.

(Remarks on code availability)

I do not have the expertise to evaluate or run the code.

Point by point response to reviewer comments

Reviewer comments: black normal font
Our responses: green normal font
Extracts from the revised manuscript: *green italic font*

General comment

We appreciate the reviewers' constructive feedback and have revised our atlas accordingly. In response to their concerns, we implemented several improvements to enhance its overall robustness and biological relevance. By revisiting the annotation process, we identified two additional cell types: (1) neutrophils, which are notoriously difficult to detect in single-cell RNA sequencing data, and (2) plasmacytoid dendritic cells. After establishing the level 1 annotations, we subsequently annotated finer grained subtypes (level 2) to inform the deconvolution of bulk RNA-seq samples.

In addition to providing our revised level 1 reference atlas, we also offer our level 2 reference atlas and associated model weights as supplementary resources. Following these changes, we recalculated all references, performed all downstream analyses again to reflect the improved biological insights. These include newly proposed marker genes for neutrophils in human atherosclerotic plaques, the vascular tissue specificity of fibromyocytes, proangiogenic endothelial cells in plaques that could be entry points for immune cells, and the role of macrophage subsets in late lesions. Together, these comprehensive revisions and responses to the reviewers' suggestions have substantially enhanced the quality and utility of the manuscript.

In the following we will address the reviewers comments individually:

Reviewer #1 (Remarks to the Author):

Traeuble et al present an integrated single-cell atlas of human atherosclerotic plaques comprising 261,747 cells from carotid, coronary, and femoral arteries. The authors manually annotated a subset of samples, applied the best-performing integration method to build the atlas, and subsequently subclustered 'level 1' cell type annotations to identify more granular cell subsets (termed 'level 2' annotations). Finally, downstream applications of the constructed atlas are demonstrated, including automated cell type annotations and bulk RNA sample deconvolution.

While this study has the potential to serve as an informative resource, the manuscript in its current format reads like a methods paper rather than offering unique biological insights into the cellular plasticity driving advanced plaque progression. A more thorough analysis of the normal versus aberrant cell states underpinning atherosclerosis would be expected from a designated atlas paper. Specific comments and suggestions of this reviewer are provided below.

We thank the reviewer for highlighting the potential of our study and insightful suggestions for improvements. We took these comments seriously and revised the atlas substantially by strengthening biological insights and focusing less on the methods. Classical analyses of normal versus aberrant cell states, as suggested by the reviewer, would be desirable but are hard to accomplish in atherosclerotic plaques due to a lack of standardized phenotyping and a general lack of “healthy” samples. For example the Hu et al. dataset is generally considered to represent control arteries by others (<https://doi.org/10.1016/j.celrep.2023.113380>). These coronary artery samples are from heart transplant patients with end stage heart failure, making the “healthy” or control phenotype tricky to compare to, as other heart failure related changes might confound the analyses. The same holds for the Tabula Sapiens patients which originate from patients with CAD or the adjacent normal samples which is why we called them “early lesions” instead of “healthy”.

We clarified this in the introduction of the manuscript with:

“While a comprehensive understanding of atherogenesis requires insights into both healthy and diseased arterial cells, obtaining truly “healthy” samples remains a significant challenge. Studies on “healthy” arteries often involve patient populations with underlying cardiovascular conditions, such as end stage heart failure or earlier stages of atherosclerosis, making it difficult to establish a definitive “normal” phenotype. To address this limitation, we focused our analysis on characterizing the cellular heterogeneity across various stages of atherosclerotic plaques. To set the stage for future comparisons, it is key to obtain a high-resolution cell type annotation of the cells of atherosclerotic plaques.”

Major Comments:

Atlas datasets

1. The authors state that current single-cell atherosclerosis atlases are limited to only carotid and coronary artery samples. However, only one femoral artery dataset is included in this atlas (GSE234077). Moreover, based on the experimental design of this dataset, single-cell suspensions of the plaques were enriched for CD45+ leukocytes for subsequent sequencing. In this regard, this dataset may not be sufficient to provide comprehensive coverage of all cell types present in plaques from femoral arteries.

We thank the reviewer for highlighting that we may have overstated the advantage of including cells from plaques of femoral origin in our study. Indeed the femoral plaque dataset is enriched for CD45+ cells which limits the coverage of other (structural) cell types. Still, it is the only available scRNAseq dataset we found for femoral plaque tissues. We believe that it is beneficial to include these femoral cells to enable harmonized annotations and to allow abundance comparisons despite not covering the full cell type spectrum.

To acknowledge this situation, we have added this limitation to the discussion section:

“We acknowledge that there are certain limitations. First, the representation of cells of different arterial origins is not balanced. In particular, femoral arteries were only sampled in a single study, where a pre-selection for CD45 positive cells was applied. Thus, structural cells in femoral arteries could not be sufficiently characterized. In general this scarcity of cells of femoral origin hinders comprehensive comparisons across different vascular beds and origins.”

2. This atlas excludes healthy samples. Can the authors further comment on this decision? Including both healthy and diseased samples in the reference atlas would allow for detection and comparison of healthy vs disease-specific cell types.

a. A random selection of healthy cells (to a pre-defined number) can be implemented to avoid over-representation of healthy cells in the integrated data.

This is an important point, which indeed requires a more nuanced motivation and discussion. As described in the first response, cells from truly “healthy” arteries are hard to obtain. Considering “non-atherosclerotic” arteries from end stage heart failure patients is currently used (<https://doi.org/10.1016/j.celrep.2023.113380>) as an approximation to control samples.

The authors of that study acknowledge the limitation of this approximation: “The limitation arises from the sourced datasets included in this meta-analysis (e.g., non-lesion samples came from patients with non-ischemic dilated cardiomyopathies, and inflammatory cell populations could be consequences of myocardial inflammation or secondary subclinical diffuse intimal thickening), and while the majority of the cell types were balanced across samples, it is difficult to disentangle biologically meaningful processes or technical factors.”

Thus in our main analysis we prefer not to compare to this sample as an approximation to „healthy“. Nevertheless, we assessed the consequences of including these cells in the atlas. In particular, we looked at the impact on the cell annotations. To this end, we retrained the whole atlas by including cells of the Hu et al dataset.

We firstly integrated several samples of the Hu et al. data set using the cell type agnostic integration method Harmony. Then, we manually annotated these samples using our default workflow and marker genes. To address the point of over-representation, we followed the suggestion of the reviewer by including a random sample selection of these “healthy” cells per cell type so the total number of “healthy” cells does not exceed 20% per cell type in the reference.

Because the reference annotations are not changing, we only evaluated the changes in the annotations of the query data (unknown cell types after level 1 corrections). We compared the predictions of the atlas once using a reference without the “healthy” cells and once using a reference with the “healthy” cells. The confusion matrix is shown in the figure below. We did not observe a major difference in cell type predictions. As in the comparison to author provided labels, the few differences that were observed lead to label switches of cell types with similar transcriptomic profiles, e.g. myeloid cells, SMC / fibroblast / fibromyocytes, TC / NK (see attached confusion matrix). Given this very minor impact of including healthy cells on cell annotations, we hence decided to exclude these samples from the reference atlas to only include cells in the atlas originating from atherosclerotic plaques.

3. Do the included single-cell atlas datasets include metadata about the severity of the disease (e.g. early vs late lesions)? If so, the manuscript would benefit from the added biological insights throughout disease progression. Further, the single-cell data could validate the results of the bulk RNA deconvolution between early and late lesions.

We concur with the reviewer that this would indeed be a very valuable addition. However, as there is no harmonized phenotypic description of disease severity within the plaque samples in the atlas, a comparison between the datasets is generally not possible with respect to disease severity. Although samples within individual data sets might be more comparable, we believe a re-analysis without integration across datasets would most likely only reproduce results already reported in the initial studies. To exploit the potential of the whole integrated atlas, we combined it with a bulk RNA-seq data set, which provides harmonized phenotyping and a large sample size to assess abundance differences between early and late lesions. Despite the lack of

harmonized phenotypic information, the atlas does include metadata for each dataset with respect to the origin of the plaque and the sorting applied prior to sequencing. Thus, we have added an abundance analysis comparing different plaque origins:

“Cell type abundance analysis using single-cell and bulk RNA sequencing

The plaque cell atlas also enables the analysis of cell type abundances across different datasets and sorting strategies. Single-cell data allows researchers to directly assess abundances or deconvolute bulk RNA-seq data for broader insights. For all compositional data analyses, the estimated abundances were centered and log ratio transformed (CLR). The atlas includes datasets from carotid, femoral, and coronary arteries, sorted by different criteria: unsorted, CD45+ cells, or T cells (see Suppl. Figure 6). This diversity facilitates sanity checks across and comparisons within these groups (see Suppl. Figure 7). The annotations of the atlas are in line with the expectations, as cells in T cell-sorted datasets are predominantly annotated as T cells and NK cells in our atlas, while CD45+ sorted datasets largely lack structural cells, which are highly abundant in unsorted datasets. Additionally, abundance patterns vary by origin. Monocytes and dendritic cells are abundant in carotid plaques, but nearly absent in the femoral ones (t-test on CLR values: $t=2.9$, $P=0.008$; $t=-2.5$, $P=0.02$). In unsorted coronary datasets, fibroblasts and endothelial cells are more prevalent, whereas T cells appear less common compared to carotid plaques. A comparison of origins using unsorted datasets is not possible, since the Wirka et al. study is the only one of coronary origin, which confounds the analysis. These findings highlight variability in cell type composition between origins, but should be interpreted cautiously due to potential biases from varying sampling and extraction methods and small numbers of tissue specimens obtained using comparable techniques. To circumvent this limitation and take advantage of larger sample sizes of bulk RNA-seq experiments, we applied deconvolution of bulk RNA samples using our atlas as a reference [...]

Cell type annotation

1. How was the level of uncertainty quantified for cell type assignment?

The scPoli framework provides uncertainties based on euclidean distance to the closest cell type prototypes in the reference, which are learned during training.

We clarified this in the manuscript by adding to the “Automatic cell type annotation” part: “*which is derived from the euclidean scaled distance to the closest cell type prototype in the reference*”.

2. Some of the marker choices for cell type annotation require additional explanation and/or review.

a. Pro-angiogenic ECs:

Why are proliferation markers or collagen/remodeling markers not included for this annotation?

Please provide references that the provided level 2 pro-angiogenic markers are indeed canonical for this subset.

b. EndoMT ECs:

The authors utilized markers characteristic of mesenchymal cells. EndoMT regulators such as Snail1 and Snail2 should also be considered.

We thank the reviewer for raising this point. Because of our revised annotations, we changed from four EC subtypes of proangiogenic, EndoMT, lymphatic and intimal, which were mapped to venous and arterial ECs, to just two cell types for which classification appears more robust. We used marker genes to annotate 'EndoMTs' with arterial origin and 'proangiogenic ECs' with venous origin. References for the new marker genes are provided in Suppl. Table 1. Regarding EndoMT marker genes: while Snail1 and Snail2 are commonly used in immunohistochemical stains, their usefulness in scRNAseq data is limited. Both genes had little to no expression in all endothelial cells investigated. The exception was one small subset in the Wirka et al dataset, which once more highlights the differences between scRNAseq studies.

We addressed this in the "Cell type abundance analysis using single-cell and bulk RNA sequencing" section by including the following paragraph:

"In the EC population, we identified one cluster expressing ACKR1, AQP1, CXCL12, FABP4, and NR2F2, which can be attributed to a proangiogenic phenotype of venous origin. The other EC subcluster expressed established EndoMT genes COL1A2 and FN1, along with co-expression of GJA4, GJA5, MECOM, and GATA2, indicating their arterial origin. Classical EndoMT regulators such as SNAI1 and SNAI2 were not expressed in any ECs, except for SNAI1 in a subset of cells from Wirka et al. (see Suppl. Figure 8 and 9). This underscores the advantage of simultaneously analyzing multiple integrated datasets and indicates that classical EndoMT regulators are not consistently expressed across all scRNA-seq studies."

3. A goal of the study is to enable accurate automatic cell type annotation of new datasets. However, the atlas does not correctly identify neutrophils, which are reported as known cell players in atherosclerosis. Can the authors comment on this limitation? Could more stringent cell annotations and/or the inclusion of additional datasets be considered to ensure these subsets are not overlooked?

We thank the reviewer for raising this important point. While neutrophils are indeed known cell players in atherosclerosis and consistently found in antibody-based detection methods, they are almost never found in scRNA-seq studies of human plaque. They are very hard to detect according to 10.1161/CIRCRESAHA.120.316903. The reason for this could be that they are rich in easily releasable ribonucleases that rapidly degrade the endogenous RNA while only expressing a few hundred genes (<https://doi.org/10.1016/j.exphem.2018.09.004>). 10xGenomics, the manufacturer of the scRNAseq technology used, also indicates that by default, the Cell Ranger software may filter out neutrophils (<https://www.10xgenomics.com/support/software/cell-ranger/latest/tutorials/cr-tutorial-neutrophils>).

We only found two occurrences, where neutrophils were detected in plaque scRNAseq data: the Bashore et al. and Mosquera et al. studies. In the Mosquera et al study, only a very small number of neutrophils were found, while Bashore et al. unraveled its own cluster based on

surface proteins (CITE-seq). Because we used this dataset to validate our automatic cell type annotations, we have access to the author provided labels. We investigated the mapping and the confusion matrix. This showed that 98% of these neutrophils were classified as monocytes by our initial atlas. Interestingly, Bashore et al. did not classify any monocytes at all in their dataset. Considering this, we revised our monocyte cluster and found a distinct subcluster. We applied differential gene expression analysis, which yielded neutrophil specific genes. Interestingly, this neutrophil cluster mainly consists of cells from one dataset, where the authors did not annotate any neutrophils in their original study. Our new neutrophil annotation was validated by mapping the Bashore et al. dataset on this newly annotated reference. Comparison with the author provided labels and the CITE-seq data demonstrated almost perfect neutrophil detection, using only mRNA markers (identified by clustering and deg analysis), instead of the surface markers used by Bashore et al.

We added one paragraph in the “Integration of public datasets into one atlas” section of the manuscript:

“Notably, we observed a distinct cluster within the monocyte population where typical monocyte markers were not expressed. This cluster consistently appeared regardless of the batch correction technique employed. It distinctly expressed neutrophil-associated genes such as NAMPT39, IFITM240, G0S241, CXCL842, NEAT141, SRGN43 and AQP944, suggesting that these cells are neutrophils. This finding is particularly interesting because neutrophils are known to be difficult to detect in scRNAseq⁴⁵. This challenge may be attributed to their high content of readily releasable ribonucleases that rapidly degrade endogenous RNA and their limited gene expression profile, consisting of only a few hundred genes⁴⁶. In line with that , 10x Genomics, the manufacturer of the scRNAseq technology used, indicates that the Cell Ranger software may, by default, filter out neutrophils. In plaque tissue, only one study²⁷ has detected neutrophils in very low numbers, and in the Bashore et al. study⁴⁷ they were identified using surface proteins.”

And two paragraphs in the “Automatic cell type annotation” section:

“Notably, while Bashore et al. had used surface protein expression to annotate neutrophils, we obtained almost perfect precision and recall for neutrophil annotations only based on transcriptomic profiles.”

“Additionally, the neutrophils, annotated based on our mRNA derived markers, express known neutrophil surface proteins CD15 and CD16. Together with the consensus to the author-provided annotation, this highlights the robustness of our proposed marker genes.”

Methodology

1. The authors identify scPoli to be the best-performing integration method in bio conservation and batch correction. Why then is Harmony used to integrate samples for the level 2 annotations (only briefly mentioned in ‘Level 2 Annotation’ subheading of Methods section)?

We thank the reviewer for highlighting the need for a more detailed motivation for this choice. To manually reannotate the cells, we wanted to use the most unbiased integration method enabling us to find level 2 cell types in a data driven way and correct for potentially misclassified cells. To achieve this, we opted to use integration methods that do not rely on prespecified cell types. Harmony performed best in this regard, by being able to identify human plaque-relevant sub cell types. After obtaining level 2 annotations, this integrated embedding was subsequently discarded and only the cell type annotation was used for further processing.

To clarify this important point, we added a paragraph in the methods section: *“Harmony was selected over other integration tools due to its effectiveness in reducing batch effects without relying on predefined cell types. This allowed for an unbiased identification of sub-cell types by minimizing the risk of clustering biases within Level 1 categories. Harmony provided an integration with good biological signal retention, which we used solely for reannotation; the integrated embedding was then discarded, and only the updated cell annotations were used in further analyses.”*

2. An integration pipeline starting with manual annotation of a subset of samples followed by scPoli integration and label transfer is employed. Is such a sequence (annotation first, integration after) a standard in single-cell analysis comprising multiple datasets that require batch correction?

According to a very recently published review article on how to build single cell atlases (<https://doi.org/10.1038/s41592-024-02532-y>), this sequence of annotation first and integration after is indeed the standard. This is because the latest and best performing integration methods such as scPoli, scGen or scANVI rely on cell type labels, which are not available without manual annotation. To still use these methods, we manually annotated several samples until we reached at least 600 cells for each cell type to have a robust reference.

To let the reader know this we changed on sentence in the “Integration of public datasets into one atlas” section to: *For this reason and following a guideline for atlas curation, we manually annotated a subset of 11 samples using a carefully curated table of human plaque-specific marker genes (see Figure 2A)*

3. While the scPower framework may serve as a useful resource for planning future experiments, this section in the text comes across quite randomly in the paper’s current format. Moreover, it is not entirely relevant to understanding the complexities underlying atherosclerosis in this manuscript.

We agree with the reviewer that strengthening the relevance of the scPower analysis for applications in atherosclerosis would improve the coherence of the paper. To address this, we are now illustrating the use of scPower with three atherosclerosis related gene sets, instead of assuming arbitrary signatures with specific expression levels. These are 1) the downstream targets of the CCL19 chemokine; 2) genes of the Interferon-Gamma (IFN- γ) signaling pathway and 3) a vascular remodelling gene set. We also slightly adapted the fold change of the most

optimistic scenario (from 2.0 to 2.5) and changed the figure in the manuscript accordingly. Consistently with the rest of the paper, the new level 1 atlas which includes the Bashore et al. dataset was used as reference.

We changed the following paragraph in the “Planning future experiments with scPower” section of the manuscript:

“[...] ScPower can estimate power for specific gene sets, which reflect the aims of the experiment and can be derived from prior knowledge. Here, we made use of this feature and evaluated the average power for three atherosclerosis-related gene sets, which are expected to be active in different cell types: (1) targets of the CCL19 chemokine, which is overexpressed in carotid plaques of symptomatic patients and its targets downstream of CCR7 are expected to be expressed in T-cells, B-cells, dendritic cells and NK cells; (2) genes of the Interferon-Gamma (IFN- γ) pathway, which is active in atherosclerotic lesions and a well known activator of macrophages; and (3) genes of the vascular remodelling pathway, which were identified through genome-wide association studies of coronary artery disease and are expected to be active in structural cells. For all combinations of these specific pathways and fold changes, we evaluated the power to identify differentially expressed genes with a total sample size of 22 individuals.

For abundant cell types the power is generally high, while lower abundance cell types show substantially less power (Figure 5). As expected, higher effect size leads to an increase in power. For detecting differential expression in the vascular remodeling pathway, structural cell types, such as smooth muscle cells (SMCs), endothelial cells (ECs), fibromyocytes, and fibroblasts, exhibit comparatively higher statistical power than other cell populations, despite their lower abundances, as the relevant genes are highly expressed in these cells. Accordingly, the IFN- γ signaling pathway demonstrates its strongest power in immune cells, including T cells, macrophages and monocytes. Within the CCL19 targets, T cells show one of the highest power estimates. Under less optimistic assumptions, the overall power for these gene sets remains below 75%, indicating that larger sample sizes are essential for achieving robust statistical detection. In case an investigation of rare cell types is desired, we recommend sorting with FACS prior to sequencing to enrich these cells enough to have sufficient power to detect differentially expressed genes. Insights like these can streamline resource allocation and inform necessary adjustments to experimental designs.”

Biological Insights/Questions

1. Are there different cell types (abundances and/or altered phenotypes of a specific subset) present between carotid, coronary, and femoral samples? There is currently no information provided from the single cell studies to understand these origin-specific plaque differences at the cellular level.

As indicated in response to comment 3, we investigated cell type differences from different origins. However, because some datasets were sorted prior to sequencing, we had to run separate abundance analyses for each sorting strategy. This led to relatively small numbers of comparable samples.

We added the following text on this analysis to the manuscript:

“Cell type abundance analysis using single-cell and bulk RNA sequencing

The plaque cell atlas also enables the analysis of cell type abundances across different datasets and sorting strategies. Single-cell data allows researchers to directly assess abundances or deconvolute bulk RNA-seq data for broader insights. For all compositional data analyses, the estimated abundances were centered and log ratio transformed (CLR). The atlas includes datasets from carotid, femoral, and coronary arteries, sorted by different criteria: unsorted, CD45+ cells, or T cells (see Suppl. Figure 6). This diversity facilitates sanity checks across and comparisons within these groups (see Suppl. Figure 7). The annotations of the atlas are in line with the expectations, as cells in T cell-sorted datasets are predominantly annotated as T cells and NK cells in our atlas, while CD45+ sorted datasets largely lack structural cells, which are highly abundant in unsorted datasets. Additionally, abundance patterns vary by origin. Monocytes and dendritic cells are abundant in carotid plaques, but nearly absent in the femoral ones (t-test on CLR values: $t=2.9$, $P=0.008$; $t=-2.5$, $P=0.02$). In unsorted coronary datasets, fibroblasts and endothelial cells are more prevalent, whereas T cells appear less common compared to carotid plaques. A comparison of origins using unsorted datasets is not possible, since the Wirka et al. study is the only one of coronary origin, which confounds the analysis. These findings highlight variability in cell type composition between origins, but should be interpreted cautiously due to potential biases from varying sampling and extraction methods and small numbers of tissue specimens obtained using comparable techniques.

To circumvent this limitation and take advantage of larger sample sizes of bulk RNA-seq experiments, we applied deconvolution of bulk RNA samples using our atlas as a reference [...]”

2. In Supplemental Figure 3, the clustering of lymphocytes in the UMAP (NK, T cells, and B cells) overlaps with the cells from femoral artery samples more so than within the myeloid populations. Is this in line with previous studies?

We greatly appreciate the reviewer’s attention to detail. The UMAP visualization may not be optimal to answer this question. Displaying so many cells on a discrete plane of pixels at the same time leads to a situation where cells can overdraw other cells. This can lead to the impression that cells of a specific cell type are more abundant. To more directly visualize cell type composition in data sets of different arterial origin, we added boxplots in the Suppl. Figure 7 (also shown below), which is based on the new level 1 atlas that includes the Bashore dataset. Because the femoral dataset (Slysz et al.) is CD45+ sorted, we expect only very few cells annotated as structural cells. This can be observed very well in the boxplot. The relative abundance for the myeloid cell types monocytes and dendritic cells are abundant in carotid plaques but nearly absent in femoral ones (t-test on CLR values: $t=2.9$, $P=0.008$; $t=-2.5$, $P=0.02$). Lymphocytes do not show significant differences in abundance between femoral and carotid origin (even though B-cells visually appear to show differences, they are not statistically significant when applying a t-test on CLR values: $t=1.5$, $P=0.16$).

3. Can the authors comment further on the finding of venous ECs in the advanced plaques? Specifically, does the mapping of pro-angiogenic ECs to venous EC origin definitively indicate neovascularization within the plaque? Or is it possible that surrounding veins were collected in addition to the analyzed arterial sample?

We thank the reviewer for asking this important question. Please note that we changed the workflow of the atlas and we have renamed these ECs to Proangiogenic ECs with venous origin. The origin of the vascular bed, from which the plaque has been removed, plays an important role in this regard. Coronary arteries, if analyzed unsorted like in the work from Wirka et al., could contain vasa vasora (with venous ECs) from the outer vessel wall as part of the adventitia. This assumption is based on carefully assessing the described procedure of extracting proximal to mid right coronary arteries from heart explants in the Wirka et al. manuscript.

Carotid plaque samples were collected during carotid endarterectomy (CEA). The surrounding tunica externa and adventitia remain in the patient, while only the innermost part of the artery (remnants of the tunica media, necrotic core and fibrous cap/neointima) is removed from the patient. Therefore no cells of surrounding veins are collected during CEA, only the innermost layers of the artery. In Supplementary Figure 8C, it is clearly visible that the vast majority (76.2%) of the pro-angiogenic ECs with venous origins come from carotid datasets (Bashore, Alsaign), which were obtained upon CEA. Hence, for the carotid data sets, these cells indeed are from within the plaques. This clearly is indicative of neovascularization in carotid plaques.

Additionally, recent studies have highlighted the recruitment of venous ECs into large arteries (Trimm & Red-Horse, Nat Rev Cardiol, 2023, doi: 10.1038/s41569-022-00770-1). This provides further evidence supporting the presence of venous-origin ECs in the plaques.

We have added this point in the description of the deconvolution results:

“While the proangiogenic ECs with venous origin are more abundant (t-test on CLR values: $P=6.9e-06$) in advanced carotid plaques, EndoMT-ECs of arterial origin present the opposing abundance pattern (t-test on CLR values: $P=2.7e-09$) in early lesion tissues compared to late plaques. The majority (76.2%) of these proangiogenic ECs with venous origin, which form the basis of the deconvolution analysis, were identified in carotid plaques, which were collected by carotid endarterectomy and thus only comprise the innermost part of the artery. Together these interesting observations confirm the notion that these EC-rich neovessels in late plaques are allegedly the entry point for immune cells that infiltrate the tissue and likely contribute to lesion progression and instability. This becomes particularly relevant when EC barrier function becomes impaired, neovessels start to leak, and intraplaque hemorrhages occur^{64,65}.”

4. Atlas papers typically require a detailed description/proposed function of each cell type comprising the atlas. A much more in-depth analysis regarding the annotated cell types and their role in disease progression is expected. If this is not the goal of the manuscript, “atlas” should be removed from the title.

We agree with the reviewer that an atlas should provide the reader with a detailed description of each cell type and its relevance for disease. A complete description of all cell types would inflate the text, since there are so many disease-relevant cell types. Therefore, we added a comprehensive new supplementary table which not only describes each cell type’s role in the disease, but also provides a more granular source list of the marker genes we used for the level 1 and level 2 cell types with a total of 46 unique references. We think this can serve as a valuable resource for the reader.

We added this in the caption of Figure 2: *“A more detailed list of references and the cell types’ role in the disease progression are provided in Suppl. Table 1.”*

Minor Comments

1. A scale should be provided for the dot plots in Figure 2 and 4 (as done for Figure 3 E).

We have added a color scale to the dot plots in Figure 2 and 4, as requested.

2. Figure 2: CD8A is listed twice in the level 2 annotation criteria for CD8 T cells.

CD8A is now listed as a level 1 marker for T cells and as a level 2 marker for CD8 T cells.

3. Page 6, 1st paragraph: “The predicted cell types in our atlas were compared to the provided labels. The precision of the annotations is 89.37% in the Pauli dataset... (see Figure 4 A-D).” This should be corrected to Figure 3. Same comment when referencing panel E later in the same paragraph.

We have amended the manuscript accordingly.

4. Page 20: "...inflammatory and foamy macrophages are substantially enriched (P-value)..."
Provide the p-value.

We have rewritten the part containing this statement, based on the results obtained with our new level 2 annotation:

"The stratified macrophage subtype abundance shows that foamy macrophages are, as expected, significantly more present (t-test on CLR values: $P=7.9e-06$) in the late compared to early lesions, reflecting the important role of macrophage infiltration and activity in late-stage atherosclerotic plaques. We did not determine any significant differences in the inflammatory, other macrophages as well as the PLIN2+/TREM1+ cluster. The latter is particularly interesting, as it is associated with vascular events, indicating that not the abundance of these macrophages, but rather their gene programs influences disease progression. Another interesting observation is the significantly higher abundance of HMOX1+ macrophages in late lesions (t-test on CLR values: $P=1.6e-04$), which may reflect the plaque's adaptation to intensified oxidative stress and changing iron-handling associated pathways. In these cells, genes such as FTL, SLC40A1, and NUPR1 are associated with iron metabolism, while antioxidative components including SELENOP and PRDX1 appear to help counterbalance the increased presence of reactive oxygen species. At the same time, the increased expression of lysosomal proteases like CTSB and CTSD, along with lysosomal machinery genes, such as LAMP2, LGMN, LIPA, and GPNMB, highlight enhanced proteolytic activity and more robust catabolic processing within these macrophage subtypes. Furthermore, the presence of genes involved in lipoprotein metabolism (APOC2, APOE, LRP1, and NPC2) suggests an active role in handling and redistributing lipids as the lesion destabilizes. Together, these molecular characteristics imply that as the plaque environment becomes more complex, these specialized macrophages respond by ramping up their iron management, antioxidative defense, and lipid-handling capacities in an attempt to preserve cellular homeostasis."

5. Higher resolution / better quality needed for Figure 6.

The revised submission contains the figure as a vector graphic. In the first submission the figure was included as a bitmap into the manuscript file. We apologize that this led to a reduced resolution of the figure.

Reviewer #2 (Remarks to the Author):

In this study, the author presents an integrated single-cell data source of 261,747 cells from human atherosclerotic plaques. Robust cell-type annotations of cells in the data source are provided and further validated independently. In the applications, the author demonstrates that the atlas enables accurate automatic cell type annotation of new datasets, optimal experimental

design, and deconvolution of bulk RNA-seq data to determine cell type proportions in human atherosclerotic lesions. However, the following concerns should be addressed before publication.

1. It is great that the author provides a comprehensive scRNA-seq data source for the study of human atherosclerotic carotid, coronary, and femoral arteries. However, a crucial concern is whether the data is comprehensive enough to contain all possible cell types or subclusters for the three types of arteries. If not, how does this issue affect the downstream analysis (i.e., cell type annotation and deconvolution)? For instance, if there are novel cell types or subclusters in new datasets, which can happen in practice, can we still use the atlas to perform cell type annotation? Similarly, can the atlas still be used for cell type deconvolution when bulk RNA-Seq data contains the cell types that are novel to the atlas?

We thank the reviewer for highlighting this issue. We are confident that our atlas is comprehensive enough for common practical use cases, as we were able to find all major cell types that were also found in earlier studies within human atherosclerotic plaques, and moreover we were able to annotate neutrophils that were overlooked by previous studies.

The general question if all possible subclusters are contained in this atlas is almost philosophical in nature, as one can never rule out that additional functionally meaningful subclusters exists, just because they were not observed or characterized in this study or in any other study before (absence of evidence is not evidence of absence). We take a pragmatic view on this question. As the field is rapidly evolving, consensus on specific sub cell types is starting to emerge. However, the level of experimental evidence for specific functional roles of these subclusters is still quite variable. In this work, we focused on subclusters that have a well described functional role, which is backed up in the literature, and that have well characterized marker genes. We performed subclustering and annotated these subclusters as level 2 cell types in our atlas, if distinct markers showed a clear signal in the data. We use these level 2 annotations in the deconvolution analysis and we provide them in the supplementary data. Of note, as the evolution of the field continues, we would expect that the level 2 annotations might need to be updated and refined more frequently when new experimental evidence for new cellular functions and corresponding markers appear, while the level 1 cell types are expected to remain stable. This is a general issue that occurs in any reference atlas. In cases where novel cell types or subclusters are encountered in new datasets, our atlas will provide a framework for identifying them. Specifically, when using reference mapping, cells corresponding to novel types or subclusters would likely receive high uncertainty scores, serving as a useful indicator for further investigation and functional characterization. Similarly, for bulk deconvolution, our atlas will not yield results for cell types or subclusters absent from the reference. This limitation underscores the importance of continuously updating the atlas to reflect new data and insights.

We are committed to improving and extending the atlas as new datasets and experimental evidence become available. Its quality, robustness, and comprehensiveness make it a solid foundation for both current research and future updates, ensuring its utility as a key resource in the field.

To reflect these considerations in the manuscript, we have expanded the discussion:

“The scPoli model, integral to this atlas, is trained on predefined cell types, limiting its predictive capacity to those included in the study. The level 1 cell type annotations are in high concordance with the author-provided annotations and represent the consensus in the field. Thus, we expect this level of annotation to remain stable. Level 2 cell type annotations are currently limited to those subtypes that have well established functional roles and established markers which clearly separate cell type clusters in our atlas. These level 2 annotations heavily depend on experimental evidence and are likely to evolve as new findings emerge. Consequently, additional experimental evidence will also uncover new functional roles for specific cell subsets. In case of a new single-cell study that includes a novel sufficiently distinct subset of cells, the model provides uncertainty metrics to flag these cells. Clusters of cells with high model uncertainty could indicate the presence of yet unidentified cell types not included in the atlas, as demonstrated with the non-plaque tissue mappings.”

2. A key benefit of the data source is to provide accurate cell type annotation, which is also the basis for cell type deconvolution with bulk RNA-seq data. A crucial concern is whether the data source can provide a superior performance in cell type annotation of new datasets, compared to the computational tools (e.g., scVI, Harmony, scPoli, scGen, scANVI, etc.) for cell type annotation, which are mentioned in the manuscript. Otherwise, there is no reason for researchers to resort to the data source for cell type annotation.

We thank the reviewer for their thoughtful comment, which reveals an ambiguity in the use of the term single cell “atlas”. We have added the following sentence to the introduction:

“These atlases usually consist of a large integrated data resource and a model trained on this data that enables annotation of the data and facilitates its re-use”.

To further address this comment, we would like to clarify that in order to obtain accurate cell annotations both components of the atlas are required. And both components should be optimized. The data resource is key to building an atlas and we have compiled all currently available data sets to ensure it is as comprehensive as possible. The model component of the atlas is intricately linked to the computational method that is used to train the model on the data resource. Many competing methods are available. Therefore we have first conducted a systematic benchmark to compare the performance of these methods on our data resource (Supplementary Figure 2). This enabled us to select the method among the computational tools (e.g., scVI, Harmony, scPoli, scGen, scANVI, etc.) that produces the model with the highest possible performance specifically for annotating cell types in our data resource, which is specific for human atherosclerotic plaques. As a result, we identified scPoli as the method that yielded the best possible annotations. Therefore, we used this method to train the model that makes up the model component of our atlas. Thus, we have optimized both components (the most comprehensive data resource possible and selected the best performing model / computational tool) and provide this for researchers to annotate new data sets.

Moreover, we have specifically evaluated the performance of the automatic cell type annotation procedure on a new data set in the section “Automatic cell type annotation”. This validated the quality of automatic cell type annotations based on our atlas. We analyzed an additional independent carotid plaque dataset of Bashore et al. consisting of more than 75k cells, of which more than 25k have additionally been profiled by CITE-seq. Our predictions achieved a precision of 94.01% and a recall of 91.74% when compared to the author-provided cell type labels, demonstrating high consensus. To corroborate the accuracy of predicted cell types independently of manual annotations, we assessed the expression levels of surface proteins measured by CITE-seq. The dot plot indicates highly cell type-specific expression of established surface protein markers for all cell types (Figure 4F).

Together these results demonstrate that researchers can rely on our atlas (data resource + model trained with scPoli) to obtain robust and accurate cell type annotations. Our approach is not a comparison of our atlas with scVI, Harmony, scPoli, scGen, or scANVI, but rather the use of scPoli as a tool to build a model that is tailored to the unique properties of our data resource. We hope this clarification addresses the reviewer’s concern and highlights the rigor of our approach.

3. On page 5, the author provides a curated table of plaque-specific marker genes for both level 1 and 2 cell types. How do the authors obtain the marker genes? The list can be incomplete. If incomplete, how are the results affected?

We thank the reviewer for their question regarding the marker genes used in our study. The marker genes were selected based on previous single-cell RNA sequencing (scRNA-seq) studies of human atherosclerotic plaques. To ensure transparency and reproducibility, we provide references for every marker gene in the newly created Supplementary Table 1, where we also include the role of each cell type in the disease.

While we acknowledge that the list may be incomplete, it reflects the current state of the literature and the existing consensus on cell type annotations in the field. Importantly, we are confident that the list is comprehensive enough to capture the key cell types and subclusters relevant to our study. This confidence is supported by the successful annotation of our cells, which we validated using independent expert-provided annotations and unbiased surface protein measurements. These validations demonstrated the robustness and accuracy of our cell type annotations.

We recognize that as the field advances and new evidence emerges, additional marker genes may be identified. However, our approach and the curated list in Supplementary Table 1 provide a solid foundation for current and future studies of human atherosclerotic plaques.

We have added considerations on the importance of the availability of high quality markers to the discussion section:

“Third, the quality of cell type annotations depends on the publicly available marker genes. While we mitigated this issue by incorporating a consensus annotation from independent experts and surface marker measurements, the experts likely relied on similar marker genes.”

Moreover, as stated in the manuscript, all 11 samples are manually annotated, but as shown in the suppl. figure 2, the comparison was conducted on a subset of manually annotated samples. I think the scalability of these methods is not a concern, and the author does not provide us with details on the selection of samples. Why was the comparison conducted on a subset of samples? How was the subset of samples selected?

We thank the reviewer for their question regarding the subset of manually annotated samples and the rationale behind their selection. Single-cell atlases rely on efficient methods to integrate heterogeneous datasets from different sources, making the choice of the best integration method crucial. To evaluate the performance of integration methods, particularly their biological conservation metrics, accurate cell type annotations are required. However, given the scale of our atlas, which includes 80 samples, manually annotating every sample was neither feasible nor robust due to variations in cell numbers across samples.

Following guidelines outlined in a recent review on building cell atlases (<https://doi.org/10.1038/s41592-024-02532-y>), we opted to manually annotate a subset of samples. This approach allowed us to identify **scPoli** as the best-performing integration method for our dataset. Once identified, we used **scPoli** to integrate all remaining samples that lacked cell type annotations.

The subset of samples was selected through a stepwise procedure. We incrementally annotated additional samples until each cell type contained at least 600 cells in total. This threshold ensures that all cell types are robustly and adequately represented. The threshold was reached after annotating 11 samples, which were not pre-selected but determined as part of this incremental process. For each sample, we applied the quality control (QC) metrics described in the methods (and in the code notebook `1_preprocessing.ipynb`) before proceeding with annotations in the `2_annotations_atlas.ipynb` code notebook. Our annotation process involved log-normalizing each sample using `scran`, excluding potential doublets, selecting highly variable genes, applying the Leiden algorithm, reducing dimensionality with PCA, and annotating clusters using our marker genes. As samples were processed individually, batch correction methods were not required in this process.

We believe this pragmatic approach strikes a balance between scalability and robustness, ensuring high-quality annotations for evaluating integration methods while enabling the integration of the remaining dataset.

4. In the level 2 annotation, Harmony was used to eliminate potential batch effects and conserve as much biological signal as possible. However, in the benchmark of different integration methods, compared with other methods, Harmony does not perform better in bio conservation and batch correction, as shown in Suppl. Figure 2. Why did the authors use Harmony, instead of other methods (e.g., scGen and scANVI) for the purpose?

We thank both reviewers for raising this point. As mentioned in the response to Reviewer 1: To manually reannotate the cells, we wanted to use the most unbiased integration method enabling us to find level 2 cell types in a data driven way and correct for potentially misclassified cells. To achieve this, we opted to use integration methods that do not rely on prespecified cell types. Harmony performed best in this regard, by being able to identify human plaque-relevant sub cell types. After obtaining level 2 annotations, this integrated embedding was subsequently discarded and only the cell type annotation was used for further processing.

To clarify this important point, we added a paragraph in the methods section: *“Harmony was selected over other integration tools due to its effectiveness in reducing batch effects without relying on predefined cell types. This allowed for an unbiased identification of sub-cell types by minimizing the risk of clustering biases within Level 1 categories. Harmony provided an integration with good biological signal retention, which we used solely for reannotation; the integrated embedding was then discarded, and only the updated cell annotations were used in further analyses.”*

5. In the manuscript, the authors provide an easy-to-use interactive Web UI to annotate new datasets including uncertainty automatically. However, the website for the tool is not provided in the code availability section. It should also be accompanied with a detailed documentation.

We thank the reviewer for pointing out the omission of the WebUI link in the code availability section. We have addressed this by adding the following information:

“The WebUI is accessible at <https://www.archmap.bio/#/genemapper/create>. To use the tool, select “Plaque” as the atlas and “scPoli” as the model. The platform will then provide detailed instructions for uploading and preparing data for annotation.”

Additionally, we have included a comprehensive user guide for the automatic mapping scripts in the GitHub repository, which provides detailed documentation on its usage. This documentation can be accessed at <https://github.com/kotr98/plaque-atlas-mapping>.

We believe these additions will ensure ease of use for researchers and improve accessibility to the tools.

Some minors:

1. There are several typos in number formatting, especially in Table 1. For example, 8.867 in Table 1 should be 8,867. Please check the manuscript to correct similar typos.

We amended the manuscript accordingly.

2. Some citations are missing. For example, there is no reference in the sentence “Subsequently, we looked for finer-grained cell types, which we refer to as “level 2”, where we focus on cell types that have distinct biological functions that have previously been described in the literature.” on page 15.

We amended the manuscript accordingly and we have compiled a list of level 1 and level 2 marker genes and literature references in Supplementary Table 1.

3. For producibility and reusage, some documentation for the GitHub repository (<https://github.com/kotr98/reproducibility-plaque-atlas>) should be provided.

We greatly appreciate the reviewer’s attention to our code. We improved the documentation on github according to the journal’s checklist for code. In particular, we have added instructions for installation and stepwise execution of the code.

Reviewer #2 (Remarks on code availability):

It seems to me that the code is reasonably detailed for reproducibility.

Point by point response to reviewer comments

Reviewer comments: black normal font
Our responses: green normal font
Extracts from the revised manuscript: *green italic font*

General comment

We thank all reviewers for their constructive feedback. In response, we have addressed all comments while also implementing several minor corrections throughout the manuscript, which we describe in detail at the end of the point by point response.

Reviewer #3 (Remarks to the Author):

Reviewer 1 raised in at least 3 different points the lack of “unique biological insights into the cellular plasticity driving advanced plaque progression”. While I appreciate the extended discussion of cell types from deconvoluted RNA-seq data, it is unclear to me whether this study provides any novel insights into mechanisms driving plaque progression.

We appreciate this feedback and would like to clarify our two main contributions:

1. Biological insights

- Fibromyocytes are vascular-specific and absent from other organs.
- Late lesions show increased pro-angiogenic endothelial cells, which likely serve as entry points for immune cells driving progression and instability.
- HMOX1⁺ macrophages play a distinct role in advanced plaques.

2. Community resource

- A curated marker-gene list for all plaque cell types, including novel neutrophil markers.
- An annotated single-cell atlas available on the CELLxGENE platform.
- Automated mapping tools provided as a Python script, Docker container, and interactive web dashboard.
- Plaque-specific priors integrated into the scPower web dashboard to guide future experiments.

What sets our atlas apart:

Unlike previous efforts (<https://doi.org/10.1016/j.celrep.2023.113380>, <https://doi.org/10.1101/2023.12.15.571796>), our atlas is built exclusively from atherosclerotic tissues—boosting the number of plaque-specific cells and improving rare-cell detection. We cover carotid, coronary, and femoral arteries, broadening anatomical scope. Finally, we release all annotations and model weights publicly, enabling immediate reuse by the research community.

Reviewer 1 raised the concern of accurate cell type annotations multiple times. I still hold this concern for the endothelial (EC) populations, where the authors have compartmentalised all ECs into “proangiogenic ECs of venous origin” and “EndoMT ECs”. I find it strange that no “normal” aortic ECs are annotated despite the atherosclerotic tissue. As EC dysfunction is a key driver of atherosclerosis development, further sub clustering of EC clusters is required to better define this population.

Thank you for the helpful comments on our endothelial cell (EC) annotations. Because the original UMAP revealed two EC clusters, we initially classified them as EndoMT and pro-angiogenic ECs on the basis of FN1/COL1A2 and ACKR1/AQP1 expression, respectively.

Old Suppl. Figure 8C: Previous Level 2 annotation of subclusters. UMAPs for the level 2 sub clustering for ECs. Cells with no clear signature were termed Undefined.

To address the reviewer’s request we sub-clustered each of these two groups. Because every EndoMT sub-cluster consistently expressed FN1, we kept the EndoMT label for all sub-clusters that originated from the original EndoMT cluster. In contrast, within the pro-angiogenic branch only one sub-cluster showed strong ACKR1 and AQP1 expression; the rest lacked both EndoMT and strong pro-angiogenic markers but expressed canonical endothelial genes such as PECAM1 and VWF, so we relabelled those cells as normal endothelial cells. The resulting dotplot is shown below:

New Suppl. Figure 9C: Level 2 annotation of subclusters. Dot plots for the level 2 sub clustering for ECs.

All downstream analyses, i.e. bulk deconvolution, cross-organ mapping and projection onto the Hu et al. dataset, were recomputed with these refined level-2 annotations, and the automated mapping pipelines were updated accordingly. The revised deconvolution reproduces our earlier findings in EndoMT and pro-angiogenic ECs, while the newly defined normal ECs show no significant difference between early and late lesions; results for macrophage subtypes and structural cells are unchanged, underscoring the robustness of the method. Incorporating the new EC subtype into the Hu et al. and Tabula Sapiens mappings likewise neither altered existing observations nor yielded additional insights.

Figure 6B: Deconvoluted abundances of the bulk samples using the finer grained level 2 cell types in our atlas. (Right) Cell type proportions; (Left) Centered and log-transformed (CLR) proportions to better highlight differences in small abundances.

To reflect these results in the manuscript, we added text to the “Cell type abundance analysis using single-cell and bulk RNA sequencing” section:

“Because the remaining ECs lacked distinct pro-angiogenic or EndoMT signatures yet expressed PECAM1 and VWF, we labeled them as ECs.”

P-values and other metrics were also adapted accordingly and we reformulated one sentence regarding the origin of the Proangiogenic ECs to make it more clear:

“One-third (31.6%) of the venous-signature, pro-angiogenic ECs in our single cell reference set originate from carotid plaques. Both the unsorted single-cell carotid profiles and the bulk RNA-seq samples were obtained by carotid endarterectomy. This procedure excises the atherosclerotic plaque together with adjacent intimal tissue while leaving the outer media and adventitia in situ, thereby ruling out contamination by endothelial cells from neighbouring veins in the deconvolution and supporting their derivation from intraplaque neovascularization.”

Figure 6 and Suppl. Figures 8C, 9C, 10, 11C, 11D, 11E were updated with the new results.

Along these lines, the authors find that “fibromyocytes” make up ~50% of cells in the plaque in their deconvolution analyses (Fig. 6A,B). Is this expected? It seems contradictory to their UMAPs of integrated data from the various scRNA-seq studies. Is this due to inaccurate markers for this population or methodological issues with the deconvolution?

We appreciate your concern regarding the high (~50%) abundance of fibromyocytes observed in our bulk deconvolution analysis. The elevated proportions of structural cells as well as macrophages, in the deconvoluted bulk samples are indeed anticipated. This discrepancy arises because single-cell RNA sequencing typically favors immune cell capture, as structural cells and macrophages are more prone to loss during library preparation. We already mentioned this in the Introduction with: “Additionally, some cell types are easier to harvest in scRNAseq, and hence more abundant in the datasets“. This was also mentioned in <https://doi.org/10.1161/CIRCRESAHA.119.315940>. In contrast, bulk RNA sequencing does not share this bias. To investigate this, we calculated the frequency per cell type in the unsorted carotid scRNAseq and the mean frequencies in the bulk deconvolution for non-diseased and diseased samples (see Table below).

It is clearly visible that lymphocytes are highly overrepresented in scRNAseq datasets (all from diseased samples) compared to bulk datasets. The same holds for monocytes, mast cells, DCs, and plasma cells. Structural cells, with exception of SMCs, and macrophages are more abundant in the late lesion samples compared to the scRNAseq. However in the early lesion bulk samples, the SMC are more abundant. The distinction between Fibromyocytes and SMCs as well as Fibromyocytes and Fibroblasts is not always very clear, because they share similar markers or are currently in transitioning phases. We saw this behavior already in the confusion

matrices of the cell type annotations, where we acknowledge this with: *“SMCs, fibroblasts and fibromyocytes are also more difficult to distinguish for the model, and they form one cluster in the UMAP”*.

Overall, we conclude that each technology can have some technical bias that can skew the absolute proportions of cell types within tissues. However, because samples from the same technology share the same bias, relative proportions obtained from comparisons within the same technologies are still faithful while comparisons across technologies have to be treated with a lot of caution and are not advised.

We added this in the discussion part in one paragraph with: *“Both bulk and scRNAseq can have technical biases that can skew the observed proportions of cell types within tissues. However, because samples from the same technology share the same bias, comparisons within technologies still reveal meaningful relative differences while comparisons across technologies have to be treated with caution and are not advised.”*

Cell type	unsorted carotid scRNAseq	early lesions bulk	late lesions bulk
T cell	0.332	0.000	0.002
Macrophage	0.181	0.193	0.338
Smooth Muscle Cell	0.149	0.194	0.056
EC	0.099	0.103	0.110
Fibroblast	0.057	0.064	0.156
Monocyte	0.050	0.002	0.007
Fibromyocyte	0.040	0.432	0.298
B cell	0.033	0.005	0.017
NK cell	0.018	0.000	0.001
Mast cell	0.016	0.000	0.004
Dendritic cell	0.016	0.002	0.004
Plasma cell	0.007	0.000	0.002
Neutrophil	0.002	0.002	0.004

The authors repeatedly point out the lack of scRNA-seq data of femoral plaques in their response. However, they have also overlooked PMID: 36547462, which analyses ~14k cells from femoral plaques. Is there a reason that the authors do not include this study, which will allow better comparisons across plaques?

Thank you for bringing attention to the dataset presented in PMID: 36547462. We are aware of this study; however, we were not able to access the full dataset. The original publication indicates “Not applicable” in its data availability section, and the supplementary materials only provide quality control metrics. Despite our efforts to contact the corresponding authors, we were not able to get a reply and access the dataset, which precluded its inclusion in our atlas. We regret any limitations this may impose on cross-plaque comparisons.

Minor points

- Methodology- reviewer point 3: while I have not read the original manuscript, I share this concern. The scPower section does come out of the blue and it is not clear how it adds to the overall story presented by the authors.

We thank the reviewer for highlighting that the scPower subsection felt disconnected.

Our overarching aim is to provide a complete single cell study workflow for atherosclerotic plaques – from study planning to data generation and, finally, automated annotation and analysis. scPower is the element that enables experimental design and we specifically parameterized it for atherosclerotic plaques. It allows cardiovascular investigators to decide, before collecting samples, how many patients and sequencing reads are needed to detect cell-type-specific gene-expression signatures of clinical interest. To make this motivation explicit we have:

1. Added a bridging paragraph at the end of the previous Results part that introduces the unmet need for prospective study design in human atherosclerosis and positions scPower as the solution: *“Having established a validated reference for plaque cell-type annotation, we next asked how the atlas could inform the prospective design of future studies. Powered sample-size estimation is a critical yet often overlooked step in single-cell experiments, particularly in human cardiovascular research where tissue availability and sequencing budgets are limited.”*
2. Renamed the subsection to *“Atlas-guided experimental design with scPower”*
3. Changed the beginning of this section to: *“Leveraging the cell-type-specific gene-expression priors contained in the plaque atlas, we applied the scPower framework to estimate the sample sizes, cell numbers and sequencing depth required to detect biologically meaningful differential-expression signatures across plaque cell populations”*

We think this way the paragraph is better integrated in the overall manuscript flow.

Reviewer #2 (Remarks to the Author):

All my previous comments have been reasonably addressed. I do not have any further comments.

We are glad to be able to address all the comments the reviewer had.

Minor changes:

In addition to the response to reviewer comments we have:

- Corrected the “HOX1+ Macrophage” label to “HMOX1+ Macrophage” in Figure 6b.
- Fixed a typographical error in Supplementary Table 1 by correcting “APOC” to “APOC1.”
- Fixed a typographical error in Table 1 and Figure 1 by correcting the number of samples.
- Identified and corrected a mistake in the normalization of the integration benchmarks. After repeating the benchmark, scPoli remained the best performing method, and

Harmony is now ranked as the second best, validating the use of Harmony in our unbiased sub-clustering approach. LIGER is now performing worse. Additionally, we have incorporated the scPoli method using a negative binomial loss function to the benchmark (scPoli_nb). We added “*scPoli (with negative binomial and mean squared error loss)*” and “[...] *scVI and LIGER performed poorly in bio conservation [...]*” to the manuscript and updated Suppl. Figure 2.

- While it's irrelevant for the integration performance, we decided to apply a comprehensive normalization across the entire atlas (i.e., all cells together) in addition to the current separately normalizing of the core atlas and the Bashore dataset before concatenating them. This additional normalization was then integrated as a new layer into the uploaded CellxGene atlas for researchers to use, while the rest remains the same.

Point by point response to reviewer comments

Reviewer comments: black normal font
Our responses: green normal font
Extracts from the revised manuscript: *green italic font*

General comment

We thank all reviewers for their constructive feedback. In response, we have addressed the last comments.

Reviewer #3 (Remarks to the Author):

The authors have addressed all my comments.

One small recommendations given the high number of fibromyocytes in the bulk versus scRNA-seq data - the authors could mention in their discussion that it may be more advantageous to evaluate atherosclerotic plaques via single Nuclei RNA-seq rather than scRNA-seq. This may preserve all cell types better.

We thank the reviewer for raising this point and added one sentence in the discussion: *"In cases where detailed characterization of structural cell populations is required, single-nucleus RNA sequencing (snRNA-seq) may be an alternative to scRNA-seq."*

Reviewer #3 (Remarks on code availability):

I do not have the expertise to evaluate or run the code.

The code is available on GitHub and Zenodo including an example of how to run the code. We also provide a docker container webUI for users with less technical expertise.